# Exceptional preservation and foot structure reveal ecological transitions and lifestyles of early theropod flyers

Michael Pittman [1] ✉, Phil R. Bell [2], Case Vincent Miller [3], Nathan J. Enriquez[2], Xiaoli Wang[4,5] ✉, Xiaoting Zheng[4,5], Leah R. Tsang[2,6], Yuen Ting Tse[1], Michael Landes[7] & Thomas G. Kaye [8]

Morphology of keratinised toe pads and foot scales, hinging of foot joints and claw shape and size all inform the grasping ability, cursoriality and feeding mode of living birds. Presented here is morphological evidence from the fossil feet of early theropod flyers. Foot soft tissues and joint articulations are qualitatively assessed using laser-stimulated fluorescence. Pedal claw shape and size are quantitatively analysed using traditional morphometrics. We interpret these foot data among existing evidence to better understand the evolutionary ecology of early theropod flyers. Jurassic flyers like *Anchiornis* and *Archaeopteryx* show adaptations suggestive of relatively ground-dwelling lifestyles. Early Cretaceous flyers then diversify into more aerial lifestyles, including generalists like *Confuciusornis* and specialists like the climbing *Fortunguavis*. Some early birds, like the Late Jurassic Berlin *Archaeopteryx* and Early Cretaceous *Sapeornis*, show complex ecologies seemingly unique among sampled modern birds. As a non-bird flyer, finding affinities of *Microraptor* to a more specialised raptorial lifestyle is unexpected. Its hawk-like characteristics are rare among known theropod flyers of the time suggesting that some non-bird flyers perform specialised roles filled by birds today. We demonstrate diverse ecological profiles among early theropod flyers, changing as flight developed, and some non-bird flyers have more complex ecological roles.

The ecology of early theropod flyers has been revealed in part by prior studies involving their anatomy, diet, locomotor abilities and habitats[1–8]. The foot anatomy of living birds varies greatly due to the diverse ecological roles they perform[9–12]. These ecological roles include leg-based launch in flying birds, perching, wading and swimming, as well as prey capture and dismemberment[13,14]. In the context of existing ecological data[1–8], we refine the ecological profiles of early theropod flyers by comparing their toe pads, foot scales, claws and joints with living birds. In particular, we combine soft tissue and joint details visible in the best-preserved specimens and under laser-stimulated fluorescence (LSF)[15], along with quantitative analysis of claw shape and size using traditional morphometrics.

The keratinised covering of the foot, referred to as the podotheca, encompasses a variety of scale types that collectively cover the fleshy

[1]School of Life Sciences, The Chinese University of Hong Kong, Shatin, Hong Kong SAR, China. [2]School of Environmental and Rural Science, University of New England, Armidale, NSW 2351, Australia. [3]Department of Earth Sciences, The University of Hong Kong, Pokfulam, Hong Kong SAR, China. [4]Institute of Geology and Paleontology, Linyi University, Linyi City, Shandong 276005, China. [5]Shandong Tianyu Museum of Nature, Pingyi, Shandong 273300, China. [6]Ornithology Collection, Australian Museum, William Street, Sydney, NSW 2010, Australia. [7]Department of Biology, University of Toronto Mississauga, 3359 Mississauga Road, Mississauga, ON L5L 1C6, Canada. [8]Foundation for Scientific Advancement, Sierra Vista, AZ 85650, USA. ✉e-mail: mpittman@cuhk.edu.hk; wangxiaoli@lyu.edu.cn

 

toe pads on the plantar foot surface[16], the lateral and dorsal surfaces of the digits, and portions of the metatarsus. In modern birds, morphological differences in the scales and toe pads have been correlated to locomotory and feeding preferences[10,11]. Specifically, flattened toe pads are found in perching and ground-dwelling birds, or non-predatory birds that use their feet for manipulation of food (e.g., parrots)[10]. 'Well-developed' pads (i.e., those with moderately convex, semi-lenticular outlines in lateral view) and/or 'protrusive' pads (i.e., those with strongly convex, semi-oval or semi-circular outlines in lateral view) are found primarily in modern birds that use their feet for hunting, as they provide additional grip[10]. Reticulate scales on the plantar surface of modern raptor feet also reflect prey preference. Sharply pointed reticulate scales (called 'spicules') are most commonly associated with the osprey (Accipitriformes: Accipitridae: *Pandion* sp.), a specialist piscivore[17], but are widespread among grasping raptors (e.g., falconids, accipitrids)[11]. Somewhat sharpened reticulate scales are also found in toucan feet (ramphastids) that are used for food manipulation and in some woodpeckers (picids)[11] that are adept vertical climbers. Like protrusive pads[10], spicules provide additional grip[18], which is critical for raptorial birds that rely on their feet for prey capture.

Two main types of toe pad arrangement are typically present in modern birds[18,19]. The arthral condition, in which the toe pad is aligned with the interphalangeal joint (Fig. 1), is characteristic of the raptorial species. The mesarthral condition, in which the toe pad is aligned with the phalanx itself (Fig. 1), is found in non-raptorial forms. These two conditions do not include pads that cover more than one entire phalanx, although the latter are also widespread among extant birds[20]. Toe pad position (i.e., mesarthral vs. arthral) in modern birds appears to be driven by prey choice and feeding behaviour rather than common ancestry[10], although there is also considerable variation within species and even individuals[20]. Arthral toe pads are found in a range of non-avialan theropods that are mostly associated with a carnivory-dominated diet (e.g., carcharodontosaurians, tyrannosauroids and dromaeosaurids[21,22]). However, since carnivory is the ancestral condition of theropods[23–25] and arthral pads have been recently identified in the ornithischian *Psittacosaurus*[26], common ancestry cannot be ruled out as a driver of their toe pad position.

Modern birds with an arthral and mesarthral arrangement both activate the tendon-locking mechanism (TLM) during grasping[27]. The TLM maintains digit flexion during perching or prey capture without additional muscular requirements, and is found in nearly all modern birds (excluding palaeognaths, e.g., emus, rheas)[27]. The pad-over-joint arthral arrangement activates the TLM more efficiently than the pad-over-phalanx mesarthral arrangement, suggesting it is a grasping

adaption[10,27]. However, it is unknown if the observed modern relationship between the TLM and toe pad alignment was also present in early flyers because the TLM is not found in palaeognaths and its fossil record is poor. Thus, arthral arrangement of the toe pads is not considered a specialised grasping adaptation in our fossil analysis.

Modern bird claw shape and size have been investigated for decades. These data have been used to study theropod palaeoecology[28] using a variety of approaches (e.g., ref. 29–31; see ref. 32, 33 for information about vertebrate claws more generally). A recent review[8] found that the most effective method was traditional morphometrics involving statistically processed measurements of toe length, relative size of the digits and claw curvature[9,34]. Specifically, traditional morphometrics was able to separate raptors that pinned prey to the ground, struck prey concussively with a closed foot, exclusively constricted prey to death, and whose talons pierced deeply into prey[9,34]. The morphometric dataset used here[12] was collected using the same measurements as[9,34] where linear measurements were taken directly and angular measurements were taken from photographs. The dataset used here[12] has a broader overall phylogenetic coverage (21 families vs 15 in ref. 9, 34), applies ecological categories to a broader range of taxa (*n* = 66 here, *n* = 39 in ref. 9, 34) and uses bone-based landmarks that better apply to fossils. The dataset also lacks measurements of non-claw phalanges compared to[9,34] because toe bones are typically disarticulated and mixed up in modern skeletal specimens which makes them difficult to identify[12]. Striking and restraining raptorial behaviour, as seen in most accipitrids and falconids, is associated with an increase in claw curvature, an enlargement of digit I, and the enlargement of an opposing digit (usually digit II)[9,12,34]. Specialised constricting raptors (i.e., raptors that have specialised pedal anatomy for constricting prey combined with a dietary niche restricted to animals small enough to be constricted) such as owls tend to have slightly increased claw curvature and claws of sub-equal size[9,12,34]. Ospreys, which specialise in piercing talons into fish, are similar to constrictors but have much more recurved talons[9,12,34]. Non-raptorial birds tend to have claws that are the same size on each toe or have an enlarged digit III[12]. Non-raptorial ground birds have claws with very little curvature[12]. Non-raptorial perching birds tend to have strongly curved claws[12] (but see ref. 35).

Modern birds may possess any combination of strongly or weakly ginglymoid foot joints (tarsometatarsal and interphalangeal articulations)[34]. Strongly ginglymoid joints are hinge-like joints that restrict the phalangeal range of motion to a sagittal plane, allowing for greater resistance to torsional loads while exerting a strong grip[34,36,37]. Weakly ginglymoid joints are much flatter, less hinge-like joints that allow a greater range of motion, including mediolateral

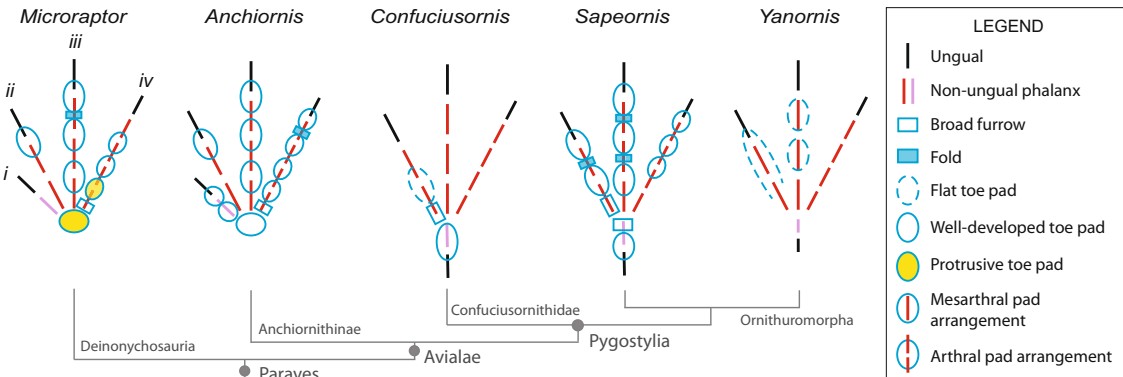

**Fig. 1 | Pedal morphology of select early theropod flyers.** Schematic of select early theropod flyers showing the proportions and arrangement of the toe bones and the morphology and arrangement of the toe pads. Missing pads (i.e., those not preserved) are not indicated. Toe pads in all taxa are arthrally arranged, except *Yanornis*, which are mesarthral in digit III. Illustrations based on *Microraptor* (STM 5-75, STM 5-109, STM 5-172), *Anchiornis* (STM 0-1, STM 0-114, STM 0-147), *Yanornis* (STM 9-531), *Sapeornis* (HGM 41HIII0405), and *Confuciusornis* (STM 13-55).

movements[34,36]. Generally, increases in joint ginglymoidy are interpreted as an increase in grip force[34,37]. Strongly ginglymoid interphalangeal joints are observed in all modern birds of prey, as well as in eudromaeosaurian dromaeosaurids[34,37]. These provide the joints with greater resistance to torsion, beneficial for hunting large struggling prey[34]. Weakly ginglymoid joints are observed in modern cursorial birds, such as ratites and emu, and in cursorial non-avialan theropods such as ornithomimosaurians[34,36,37]. Weakly ginglymoid interphalangeal joints on digit III are considered highly indicative of cursoriality, as the central toe of a running animal is expected to experience low torsional stress[34,36]. Digits II and IV may be either ginglymoid or weakly ginglymoid in cursorial taxa since they resist lateral movement and thus experience more torsional strain[34,36].

Here we apply this foot-based ecomorphological framework from modern birds to exceptionally well-preserved specimens of the early theropod flyers *Ambopteryx* (gliding only[6]), *Anchiornis*, *Archaeopteryx*, *Confuciusornis*, *Fortunguavis*, *Microraptor*, *Sapeornis* and *Yanornis*[4]. We do so on the basis that the study taxa are the closest fossil relatives of modern birds and have feet that share morphological and functional similarities with modern birds[9,21,22,28–31,34,37,38]. Some modern bird lifestyles require specific adaptations, e.g., modern perching and raptorial lifestyles require one or more opposable digits. There are special cases where lines of evidence may point to such modern lifestyles but associated specialisations are either absent or less developed. The implications of these special cases are discussed as these may shed valuable light on the evolution of modern lifestyles. Modern crocodylian feet are not useful comparisons to the study taxa as they are morphologically and functionally distinct in being plantigrade, often webbed and specialised for a semi-aquatic lifestyle[39]. We interpret our fossil foot data in the context of existing ecological data for early theropod flyers to refine our understanding of the ecological profiles present at the origins of theropod flight.

## Results

Over 1000 fossils of a phylogenetically broad range of early theropod flyers were studied under LSF[40] at the Shandong Tianyu Museum of Nature in Shandong Province, China. This collection revealed 12 specimens with toe pads, foot scales and claws that were either partially or completely preserved and belong to the early flyers *Anchiornis*, *Confuciusornis*, *Sapeornis*, *Yanornis* and *Microraptor*. This subset samples anchiornithids, early-diverging pygostylians, early-diverging ornithuromorphs and microraptorines. However, to better sample the earliest avialan flyers, the Berlin and Thermopolis *Archaeopteryx* specimens were studied first-hand in Germany and the United States, respectively, although these did not yield pedal soft tissues under LSF. To further increase the phylogenetic breadth of our analyses, specimens of *Ambopteryx* and *Fortunguavis* were also included using claw data from the literature and published photographs, even though they lacked well-preserved pedal soft tissues. *Ambopteryx* was chosen as an example of a non-paravian theropod flyer (gliding only[6]) whilst *Fortunguavis* was chosen to represent Enantiornithes. Definitions of soft tissue features are provided in the Methods section and in Supplementary Fig. 1. In this section, we present morphological descriptions of skeletal and soft tissue pedal anatomy (Figs. 1–6) as well as traditional morphometrics of pedal claws that are interpreted via principal component analysis (PCA) and linear discriminant analysis (LDA) (Figs. 7 and 8 and Tables 1 and 2).

### Morphological description: *Ambopteryx*

*Ambopteryx* (IVPP V24192) may have weakly ginglymoid interphalangeal articulations on the distal end of digits III-2 and IV-3[41]. However, this observation is uncertain due to the poor preservation of the phalanges. The articulation surfaces of the remaining phalanges and of the distal facets of the tarsometatarsus also could not be identified due to their poor preservation.

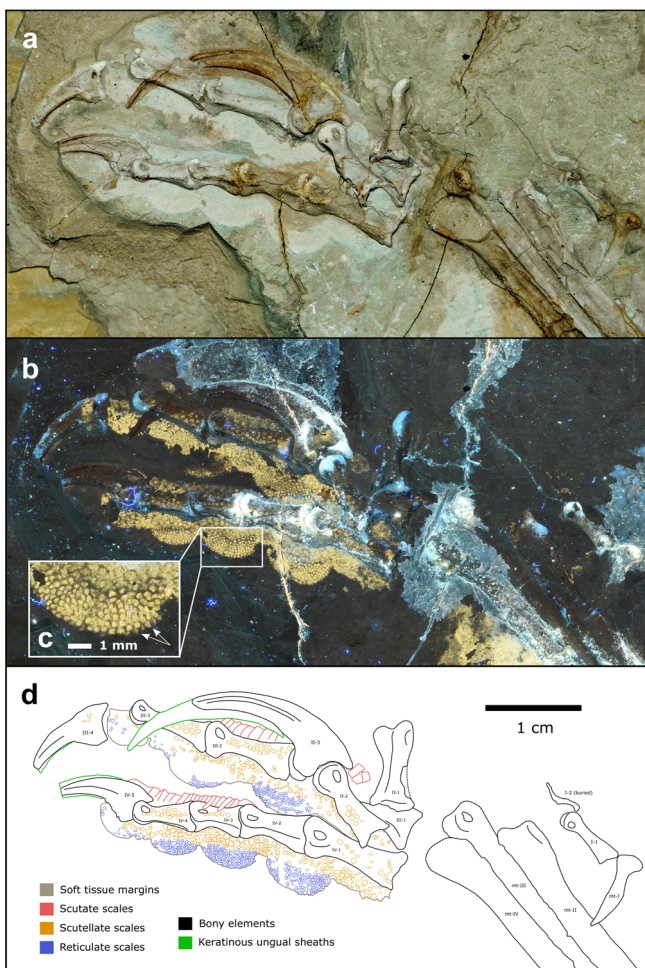

**Fig. 2 | Near-complete, semi-articulated right pes of *Microraptor* STM 5-109.** Fossil under **a** white light and **b** laser-stimulated fluorescence. **c** Close-up of reticulate and scutellate scales on digit IV, with arrows indicating exemplary spiculate reticulate scales. **d** Interpretive drawing, where solid lines indicate preserved structural boundaries, and dashed lines represent assumed or completed boundaries. Note the arthral arrangement of toe pads in which the interpad creases do not align with the phalangeal joints.

### Morphological description: *Microraptor*

The digital pads in *Microraptor* (STM 5-75, STM 5-109, STM 5-172) are arthrally arranged (Figs. 1 and 2). Due to inconsistent preservation, only a single pad—the claw pad, which covers the claw tubercle—is visible on digit II of any specimen within our sample (STM 5-75). This pad is partially preserved but appears well-developed. Digit III preserves three well-developed pads (including the claw pad), with a broad fold between the first and second pad, and a smaller fold between the second and third pad. On digit IV, there are four pads, including the claw pad. In STM 5-109, the first pad (spanning the joint between phalanges IV-1 and IV-2) is protrusive, semi-oval in outline, and separated from the second pad by a broad fold. Furrows separate the remaining pads. In STM 5-75, where it is best preserved, the tarsal pad is protrusive and widely separated by a furrow from the first pad of digit IV.

Based on the most wholly preserved specimen, STM 5-109, three distinct scale morphologies are present: (1) sub-rectangular, polarised, scutate scales on the dorsal part of each digit (~0.32–1.05 mm anteroposterior length); (2) polygonal, subrounded and irregular, non-imbricating scutellate scales on the lateral surfaces of the phalanges and dorsolateral surfaces of the digital pads (~0.13–0.88 mm diameter), and (3) minute polygonal, subrounded and irregular, non-

**Table 1 | Posterior probabilities predicting non-avian paravian ecology by LDA of TM data on modern avian claws**

| | Specimen | Constrict | Ground | Perch | Restraint | Scavenge | Strike |
|---|---|---|---|---|---|---|---|
| **Digit I Included** | *Ambopteryx* IVPP V24192 | 6.29E-01 | 1.23E-01 | 2.19E-02 | 7.56E-03 | 1.83E-01 | 3.53E-02 |
| | *Anchiornis* STM 0-1 | 1.08E-02 | 9.19E-01 | 3.21E-06 | 1.51E-02 | 1.44E-08 | 5.49E-02 |
| | *Anchiornis* STM 0-114 | 2.41E-01 | 7.59E-01 | 4.82E-06 | 7.70E-07 | 1.97E-05 | 8.44E-05 |
| | *Archaeopteryx* MB.Av.101 | 2.63E-01 | 5.96E-02 | 9.57E-03 | 1.19E-02 | 6.53E-01 | 3.52E-03 |
| | *Archaeopteryx* WDC-CSG-100 | 8.32E-01 | 3.28E-04 | 2.27E-03 | 5.36E-03 | 1.11E-01 | 4.92E-02 |
| | *Confuciusornis* IVPP V13156 | 9.34E-01 | 4.76E-02 | 1.60E-02 | 8.37E-05 | 3.21E-04 | 1.91E-03 |
| | *Confuciusornis* STM 13-55 | 9.86E-01 | 5.29E-03 | 9.64E-04 | 1.16E-04 | 1.58E-03 | 5.99E-03 |
| | *Fortunguavis* IVPP V24192 | 3.44E-02 | 4.18E-04 | 5.41E-01 | 2.23E-01 | 9.76E-05 | 2.01E-01 |
| | *Sapeornis* HGM 41HIII0405 | 2.61E-02 | 3.83E-06 | 6.70E-01 | 7.34E-02 | 1.81E-03 | 2.28E-01 |
| | *Yanornis* IVPP V13558 | 9.94E-01 | 5.86E-03 | 5.83E-06 | 2.45E-06 | 1.12E-04 | 1.34E-04 |
| **Digit I Excluded** | *Ambopteryx* IVPP V24192 | 3.85E-01 | 1.34E-01 | 3.26E-02 | 1.18E-01 | 1.36E-01 | 1.95E-01 |
| | *Anchiornis* STM 0-1 | 1.62E-01 | 8.20E-01 | 6.19E-06 | 1.08E-02 | 1.68E-05 | 6.54E-03 |
| | *Anchiornis* STM 0-114 | 4.04E-02 | 9.58E-01 | 1.94E-05 | 1.67E-04 | 8.48E-06 | 1.48E-03 |
| | *Anchiornis* STM 0-147 | 4.27E-01 | 7.70E-01 | 2.53E-03 | 7.93E-03 | 1.24E-02 | 4.02E-01 |
| | *Archaeopteryx* MB.Av.101 | 7.76E-02 | 8.11E-02 | 1.88E-02 | 6.47E-01 | 1.14E-01 | 6.09E-02 |
| | *Archaeopteryx* WDC-CSG-100 | 1.19E-01 | 2.67E-04 | 3.87E-03 | 3.38E-01 | 1.94E-02 | 5.19E-01 |
| | *Confuciusornis* IVPP V13156 | 8.76E-01 | 7.02E-02 | 4.02E-02 | 2.04E-03 | 6.95E-04 | 1.13E-02 |
| | *Confuciusornis* STM 13-55 | 7.73E-01 | 1.70E-02 | 6.57E-03 | 2.11E-02 | 2.05E-03 | 1.81E-01 |
| | *Fortunguavis* IVPP V24192 | 4.45E-01 | 3.68E-04 | 4.37E-01 | 6.41E-02 | 1.10E-02 | 4.19E-02 |
| | *Microraptor* STM 5-75 | 4.63E-02 | 1.66E-03 | 5.01E-04 | 8.04E-01 | 1.77E-03 | 1.46E-01 |
| | *Microraptor* STM 5-109 | 2.44E-02 | 1.82E-01 | 1.01E-05 | 3.70E-01 | 5.95E-04 | 4.23E-01 |
| | *Microraptor* STM 5-172 | 3.87E-01 | 4.65E-02 | 1.49E-03 | 3.37E-01 | 7.60E-04 | 2.27E-01 |
| | *Sapeornis* HGM 41HIII0405 | 1.54E-01 | 4.11E-06 | 5.91E-01 | 6.08E-02 | 2.95E-02 | 1.64E-01 |
| | *Yanornis* IVPP V13558 | 8.83E-01 | 6.84E-02 | 1.90E-04 | 7.33E-03 | 1.14E-04 | 4.06E-02 |
| **Key** | | Least Likely | | | | | Most Likely |

Values with darker backgrounds are more likely, values with lighter backgrounds are less likely (see the 'Key' row). *Ambopteryx*, *Confuciusornis*, and *Yanornis* all have claws most similar to specialised constricting raptors, followed by ground birds. Ecology is uncertain for *Anchiornis*, with different specimens showing affinities for ground bird, specialised constricting raptorial, or striking raptorial lifestyles. *Archaeopteryx* ecology is similarly uncertain, with different specimens showing an affinity for scavenging and striking raptorial use of the pes. *Fortunguavis* shows affinity with perching birds, though with digit I excluded it appears more similar to specialised constricting raptors. *Microraptor* shows a consistent affinity for restraint raptorial style followed distantly by striking raptorial style. *Sapeornis* shows a strong affinity for a perching non-raptorial lifestyle, followed by striking and specialised constricting raptorial styles. The probability of a piercing ecology is not given as only one bird, *Pandion haliaetus*, represented this category in our dataset.

**Table 2 | p values from phylogenetic honest significant differences (HSD) test, testing whether modern bird claws with different ecological groupings have different shapes, via traditional morphometrics when digit I is excluded**

| | Constrict | Ground | Perch | Restraint | Scavenge | Strike |
|---|---|---|---|---|---|---|
| Constrict | | 6.58E–01 | 1.85E–01 | 3. 82E–01 | 4.77E–01 | 7. 73E–01 |
| Ground | 6.58E–01 | | 4.00E–03** | 2.20E–02* | 1.52E–01 | 3.08E–01 |
| Perch | 1.85E–01 | 4.00E–03** | | 8.00E–01 | 8.49E–01 | 2.11E–01 |
| Restraint | 3.82E–01 | 2.50E–02* | 8.00E–01 | | 9.95E–01 | 5.13E–01 |
| Scavenge | 4.77E–01 | 1.52E–01 | 8.49E–01 | 9.95E–01 | | 7.53E–01 |
| Strike | 7.73E–01 | 3.08E–01 | 2.11E–01 | 5.13E–01 | 7.53E–01 | |

Table S1 in ref. [12] provides data for when digit I information is included; inclusion of digit I does not affect which groups are significantly different. *p* values are indicated with one asterisk (*) for significance at the 0.05 level and two (**) at the 0.01 level. Note that the pairwise() function in RRPP[48] places a lower limit on the returned *p* value, so *p* values reported as 1.00E–03 may be more significant. Adjustments are made for multiple comparisons in HSD.

imbricating reticulate scales that cover the undersides of the digital pads (~0.10–0.55 mm diameter). Where exposed in lateral view in STM 5–109 (i.e., along the ventral margins), the reticulate scales are strongly convex and resemble spicules on at least the second pad of digit IV (Fig. 2c).

Digits II, III, and IV of *Microraptor* all have strongly ginglymoid interphalangeal joints (e.g., STM 5–109, STM 5–172) (Fig. 2a). The joints are particularly well preserved in STM 5–109. Although sagittal furrows at the distal end of the phalanges are not completely observable, the hinge-like distal articulation facets are distinct in almost all of the

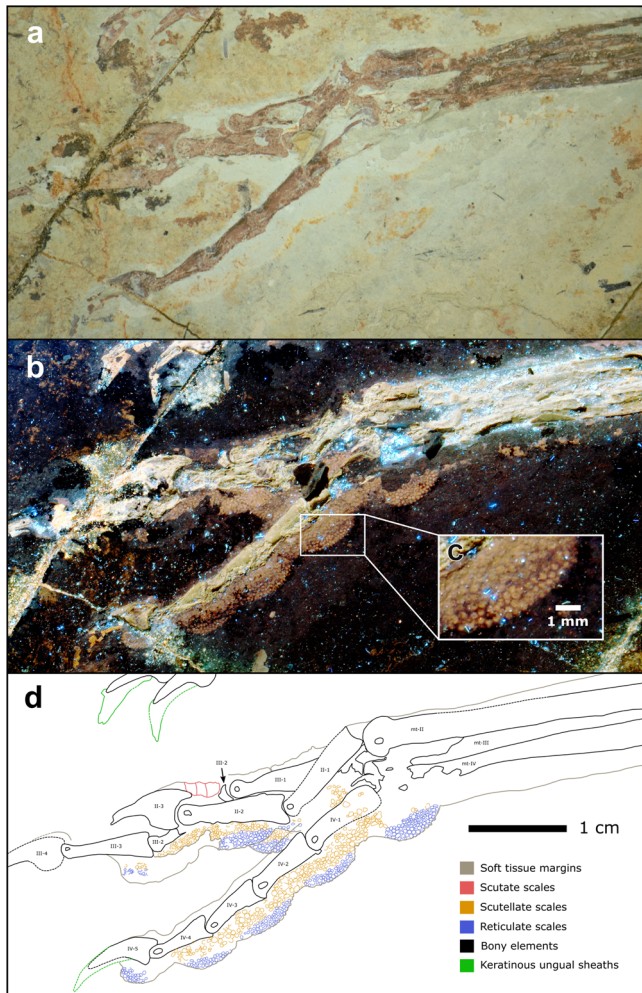

**Fig. 3 | Near-complete, articulated right pes of *Anchiornis* STM 0-147.** Fossil under **a** white light and **b** laser-stimulated fluorescence. **c** Close-up of reticulate and scutellate scales on digit IV. **d** Interpretive drawing, where solid lines indicate preserved structural boundaries, and dashed lines represent assumed or completed boundaries.

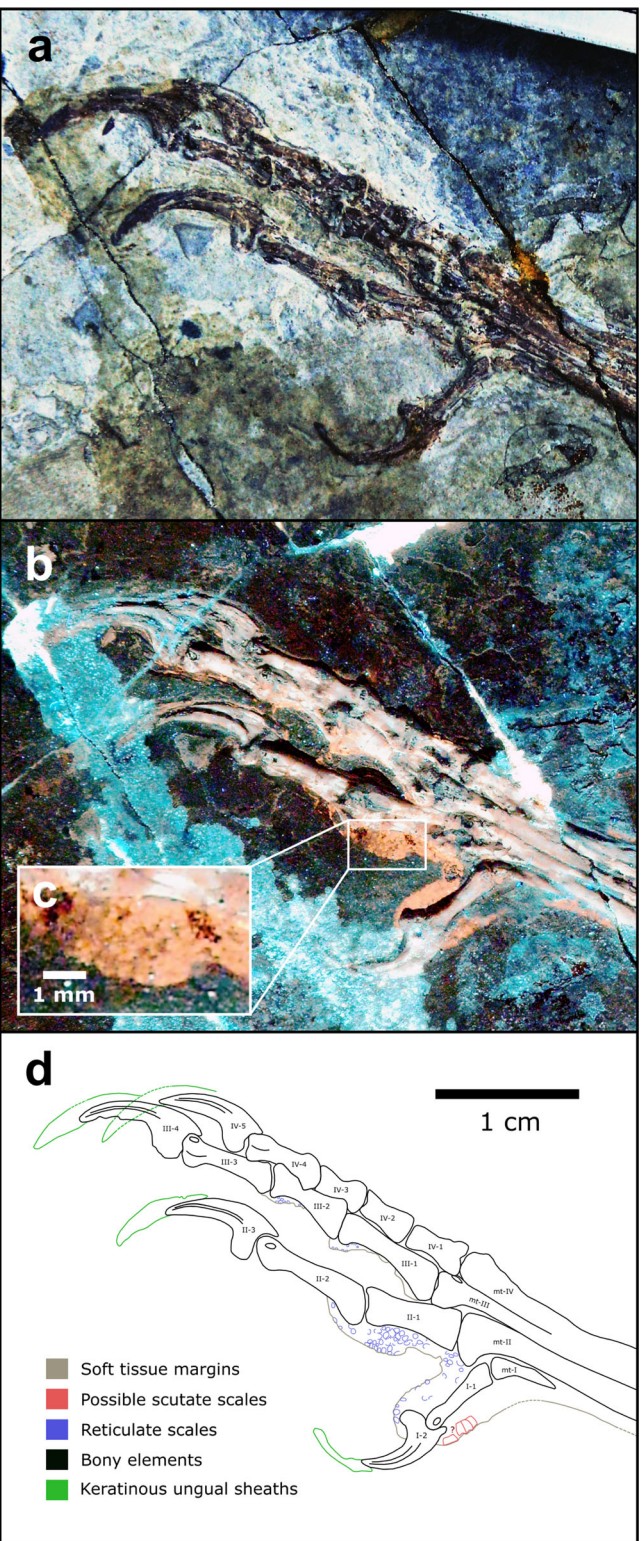

**Fig. 4 | Complete, articulated right pes of *Confuciusornis* STM 13-55.** Fossil under **a** white light and **b** laser-stimulated fluorescence. **c** Close-up of reticulate scales on digit II. **d** Interpretive drawing, where solid lines indicate preserved structural boundaries, and dashed lines represent assumed or completed boundaries.

phalanges. The concave proximal ends of the phalanges are also easily observed in lateral view. Distal facets of metatarsal II and metatarsal III in STM 5–109 are moderately ginglymoid. Joint articulations are unclear in *Microraptor* STM 5–75 due to poor preservation, but presumably resembled those of the other specimens in life.

### Morphological description: *Anchiornis*

The toe pads of *Anchiornis* (STM 0–1, STM 0–7, STM 0–114, STM 0–125, STM 0–144, STM 0–147) are arthrally arranged (Figs. 1 and 3). The hallux preserves a well-developed claw pad and two smaller pads along the length of the first phalanx. Only the large, well-developed claw pad of digit II is visible in any specimen (STM 0–114). Three well-developed toe pads are present on digit III, although they are not all represented in any single specimen; furrows/folds cannot be discerned as a result. On digit IV, there are four relatively low pads that are semi-lenticular in outline, and therefore are less well-developed than in *Microraptor*. A fold is present between the penultimate pad and the claw pad (STM 0–1, STM 0–114), whereas furrows separate the remaining pads. Unlike in *Microraptor*, the tarsal pad (STM 0–144, STM 0–147) is well-developed (not protrusive) and separated from the similarly sized first pad of digit IV by a wide furrow. All pads, including the tarsal pad, are semi-lenticular in the lateral aspect. Minute reticulate scales cover the

ventral surfaces of the digital and tarsal pads in STM 0–147 (~0.14–0.48 mm diameter on the digital pads, up to ~0.56 mm diameter on the tarsal pad). These reticulate scales are subrounded in plantar view, uniformly distributed, and typically globose (but have lower relief than in *Microraptor*), although in part of STM 0–114, most

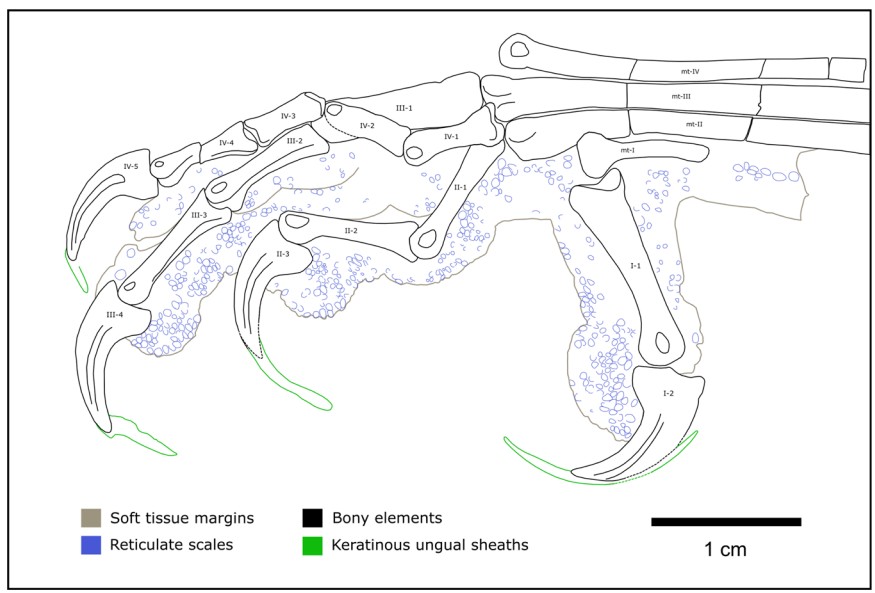

**Fig. 5 | Complete, articulated left pes of *Sapeornis* 41HIII0405.** Interpretive drawing, where solid lines indicate preserved structural boundaries, and dashed lines represent assumed or completed boundaries. Based on a photograph by S. Abramowicz in ref. 106.

noticeably on digit IV, they appear to form sharp spicules. Scutellate scales (-0.19–0.83 mm diameter) cover the lateral and dorsolateral portions of the digits in STM 0-147, and are most frequently subrounded, or less commonly polygonal-to-irregular. Probable scutate scales are only sporadically visible in STM 0-147, where they form a faint row of polarised scales dorsal to some of the phalanges (-0.96–1.40 mm anteroposterior length).

In most of the *Anchiornis* specimens we studied (STM 0-1, STM 0-125, STM 0-147) digits II, III, and IV have weakly ginglymoid interphalangeal joints (Fig. 3a). The two exceptions include STM 0-7 and STM 0-114, in which digit IV of STM 0-7 has ginglymoid interphalangeal joints, and digit III-1 of STM 0-114 has a distal end that appears ginglymoid. The distal tarsometatarsal facets are also weakly ginglymoid (STM 0-1, STM 0-7, STM 0-144).

**Morphological description: *Archaeopteryx***
In the Berlin specimen of *Archaeopteryx* (MB.Av.101)[42], the interphalangeal joints are all weakly ginglymoid. The distal facets of the tarsometatarsus are not observable in this specimen. The Thermopolis specimen of *Archaeopteryx* (WDC-CSG-100)[43] has ginglymoid joints in digit II-1 of the right pes and digit III-1 of the left pes, in which the hingelike distal articulation can be observed clearly. The interphalangeal joints of digit IV and digit III-2 of the right pes appear to be weakly ginglymoid. The distal tarsometatarsal articulations are weakly ginglymoid on both left and right pes. No observable podotheca is preserved for *Archaeopteryx*.

**Morphological description: *Confuciusornis***
We imaged over 500 specimens of *Confuciusornis* using LSF at the STM and IVPP. We identified the best pedal soft tissues in two specimens, STM 13-55 and IVPP V13156. STM 13-55 is a complete, articulated female individual based on the absence of ornamental retrices on the tail[44]. The pedes are exposed in plantar view, although only the left foot preserves notable details of the podotheca (Fig. 4). Digit I is well preserved and shrouded by a single pad that encloses the first phalanx and the flexor tubercle of the claw. The first toe pad of digit II is the only additional pad preserved on the specimen and has a similar level of development to the toe pads of *Anchiornis*: the pad is relatively flat, semi-lenticular in outline, spans the joint between the first and second phalanx (i.e., exhibits arthral arrangement; Figs. 1 and 4), and is

separated from digit I by a broad furrow. Reticulate scales on the digits are small (-0.11–0.62 mm in diameter) but not clearly visible due to imperfect preservation. The ventral margins of the pads reveal reticulate scales with low domical (globate) shapes. Incompletely preserved scutate scales are present on the dorsal surface of digit I and on the proximal end of the metatarsus. A previously identified tarsal pad (Fig. 4c of ref. 38) on the left foot of a different specimen (IVPP V13156) could not be verified. Inspection of the published figures show that the margins are not well preserved enough to identify this as a toe pad or as a possible web between digits I and II. These authors[38] also reported the presence of large phalangeal pads and smaller 'interphalangeal pads' (=folds) on the digits; however, we also could not identify these on the available figures.

Only one specimen described above, *Confuciusornis* STM 13-55, has clear preservation of the pedal joints. The interphalangeal joints of digits II, III, and IV are ginglymoid (Fig. 4a, b). From the right pes, hingelike distal articulation is particularly visible on digit II-2, digit III-1, and digit III-3. In the tarsometatarsus, the distal facet of metatarsal III is ginglymous although the facet is unclear in metatarsal I, II, and IV.

**Morphological description: *Sapeornis***
The podotheca of *Sapeornis* is well preserved on the feet of specimen 41HIII0405 (Fig. 5). All digital pads, including that on the hallux, are well-developed and arthrally arranged (Figs. 1 and 5). Based on the left pes, digit I has only a single (claw) pad, with a wide furrow that extends most of the length of phalanx I-1. Digit II of the left pes bears two pads, including the claw pad, separated by a fold. A wide furrow appears to be present between the proximal pad of digit II and the furrow on digit I. As seen on the right pes, digit III bears three pads including the claw pad, each separated by a fold. Only the claw pad is unambiguously visible on digit IV of the left pes, although the outline of two additional pads on digit IV of the left pes may be superimposed over part of the adjacent digit III. The right pes more clearly shows at least three visible pads on digit IV. The tarsal pad is not visible on either pes, although we cannot rule out that it hasn't been obscured by the hallux or another part of the foot. Only a single type of scale is present: subcircular-to-irregular, globose reticulate scales are distributed along the underside of all four digits (as preserved) and along the dorsal margin on digit I, where they are less well-defined. These reticulate scales measure -0.15–0.73 mm in diameter on the digits, and up to 0.94 mm in

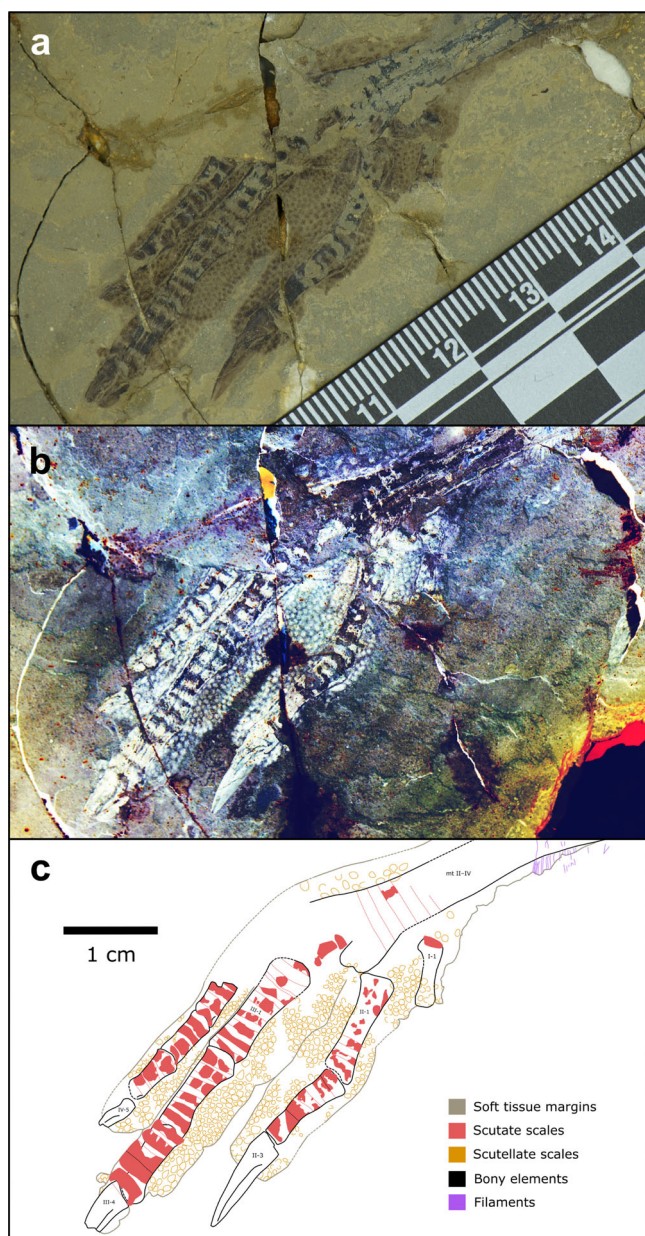

**Fig. 6 | Near-complete, articulated right pes of *Yanornis* STM 9-531.** Fossil under **a** white light and **b** laser-stimulated fluorescence. **c** Interpretive drawing, where solid lines indicate preserved structural boundaries, and dashed lines represent assumed or completed boundaries.

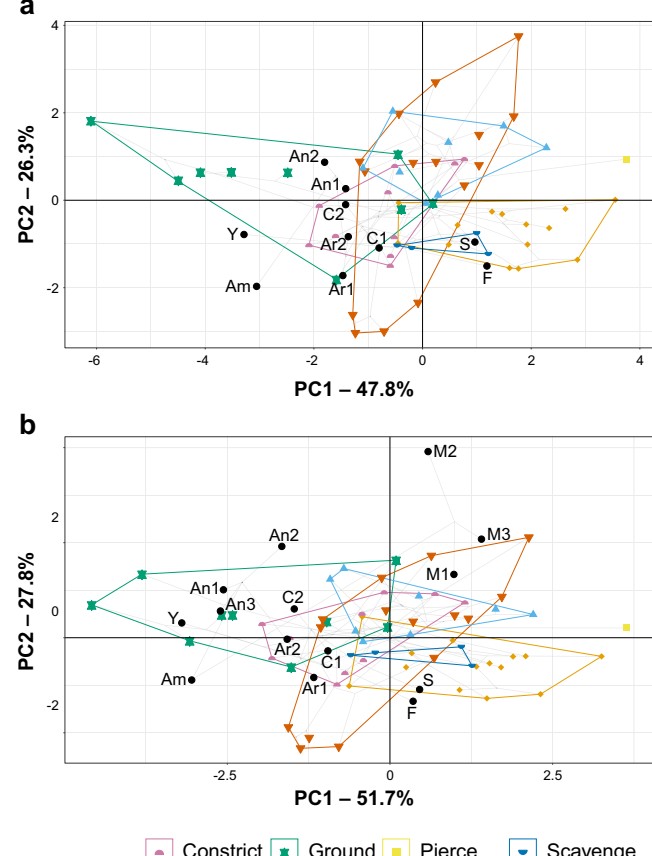

**Fig. 7 | PCA phylomorphospace of modern avian and early theropod flyer claws, based on traditional morphometrics, grouped by ecological category.** Datasets **a** include and **b** exclude measurements of digit I, as most of the fossil paravians in this study do not preserve digit I. In both graphs, claw curvature increases across PC1 and claw size relative to digit III increases across PC2. See Fig. 8 for precise character loadings. Taxon abbreviations: Am *Ambopteryx* IVPP V24192, An1 *Anchiornis* STM 0-1, An2 *Anchiornis* STM 0-114, An3 *Anchiornis* STM 0-147, Ar1 *Archaeopteryx* MB.Av.101, Ar2 *Archaeopteryx* WDC-CSG-100, C1 *Confuciusornis* IVPP V13156, C2 *Confuciusornis* STM 13-55, F *Fortunguavis* IVPP V24192, M1 *Microraptor* STM 5-75, M2 *Microraptor* STM 5-109, M3 *Microraptor* STM 5-172, S *Sapeornis* HGM 41HIII0405, Y *Yanornis* IVPP V13558.

diameter on the ventral surface of the metatarsus. Spicules and other scale types are absent.

The interphalangeal joints of digits II, III, and IV are strongly ginglymoid in *Sapeornis* specimen 41HIII0405 (Fig. 5). The proximal end of the phalanges are all visibly concave on the left pes. The hinge-like distal end of the phalanges are visible, and ginglymoidy is prominent in phalanges II-1 and III-2 where the round distal ends fit into the concavity of the proximal end of the adjacent phalanges. The distal articulations of metatarsal II and metatarsal III are ginglymoid and distal metatarsal IV is weakly ginglymoid in the left pes.

### Morphological description: *Fortunguavis*
*Fortunguavis* (IVPP V18631) has a mixture of fully and weakly ginglymoid joints[45]. Although the preservation of the distal end of digit II-1 makes it difficult to observe the joint surface, the sagittal ridge on the

proximal end of digit II-2 suggests a ginglymoid articulation. A hinge-like sagittal furrow is present at the distal end of all of the digit III non-ungual phalanges. The proximal end of digit III-3 appears concave with an observable sagittal ridge, whereas the proximal ends of digit III-1 and III-2 are less concave. Thus, digit III has a combination of ginglymoid and weakly ginglymoid articulations. Most phalanges of digit IV are too crushed to observe articulations, though IV-4 does appear fully ginglymoid at both ends. We are unable to determine if the distal facets of metatarsals I or II are ginglymoid. Metatarsal III has a weakly ginglymoid distal facet, and metatarsal IV has a ginglymoid distal facet. No observable podotheca is preserved for *Fortunguavis*.

### Morphological description: *Yanornis*
The feet of *Yanornis* STM 9-531 are exposed in dorsal view, therefore it is not possible to determine the pad shape (Fig. 6). Nevertheless, shallow invaginations in the medial margin of the integument along the length of digit III confirm that the toe pads were mesarthrally arranged within this particular digit, which contrasts with the other taxa in this study (Fig. 1). However, digit II appears to show arthral arrangement of its single distinguishable toe pad. Other details are

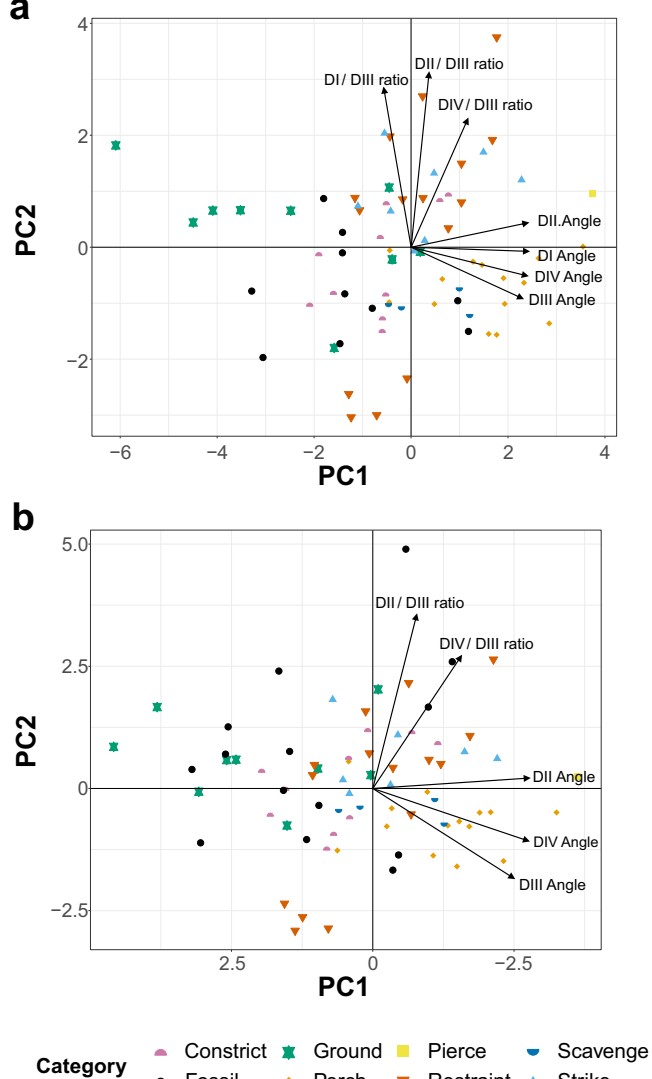

**a**

**b**

**Category**
- Constrict
- Fossil
- Ground
- Perch
- Pierce
- Restraint
- Scavenge
- Strike

**Fig. 8 | Plot of character weightings for the graphs in Fig. 7.** Plots are provided for datasets with **a** digit I included and **b** excluded. In both plots, PC1 is dominated by claw curvature where increased curvature corresponds to increasing PC1 values, and PC2 is dominated by increasing ratios of arc lengths of digits I, II, and IV relative to digit III.

difficult to confirm as digits II and III and digits III and IV are closely appressed for most of their length; only the claw of digit IV and phalanges III-3 and III-4 (claw) of digit III are free. Two scale types are visible on the pes: scutellate scales (-0.23–1.07 mm diameter) covering most of the dorsal surface of the pes, and scutate scales covering the dorsal surfaces of the phalanges of digits II–IV, and the distal one-third of the metatarsals (-0.57–1.87 mm anteroposterior length). Definitive reticulate scales cannot be seen as preserved in the dorsal aspect.

All *Yanornis* specimens studied (STM 9-531 [Fig. 6a] and IVPP V13358[46] which lacks a podotheca) have weakly ginglymoid interphalangeal joints. Weak hinge-like distal ends of phalanges are visible in IVPP V13358. The proximal ends of the phalanges are only weakly concave. Morphologies of the distal facets of the tarsometatarsus in both STM 9-531 and IVPP V13358 are not visible.

**Traditional morphometrics**
Principal component analysis (PCA) and linear discriminant analysis (LDA) plots of traditional morphometrics (TM) data are provided in

Fig. 7, with character weights plotted in Fig. 8. Discriminant predictions for LDA are provided in Table 1. In the PCA, PC1 represents claw curvature and PC2 represents the ratio in size between the digits of the foot. In the LDA, claw curvature has little effect on separation (though its effect generally increases in LDs that are less influential overall), with the ratio of digits I and II to digit III opposed to the effect of digit IV. See ref. 12 for a discussion of the trends for modern groups in these data. As digit I is not preserved in several fossil specimens, data are presented that include and exclude digit I, so that all of the study specimens could be included.

PC1 and PC2 (Fig. 7) explain 74.1% of the total variance when all digits are included and 79.5% with digit I excluded. *Ambopteryx* (IVPP V24192) occupies a unique region of the morphospace, plotting on PC1 similar to ground birds and on PC2 similar to perching birds. One specimen of *Anchiornis* (STM 0-114) also inhabits a unique region of the morphospace, with highly enlarged unguals I and II and extremely low claw curvature (with and without digit I included), while the remaining two specimens (STM 0-1, STM 0-147) plot among ground-dwelling birds. Two specimens of *Microraptor* (STM 5-75, STM 5-172) plot among restraining raptors, with the third (STM 5-109) in an otherwise uninhabited region of the morphospace representing a uniquely large digit I and II relative to digit III. *Archaeopteryx* (MB.Av.101, WDC-CSG-100) and *Confuciusornis* plot in the intermediary region between ground birds and specialised constricting raptors. With digit I included, all specimens show greater raptorial affinity, except *Archaeopteryx* specimen MB.Av.101, which shows greater ground bird and shrike affinities. Regardless of the inclusion of digit I, *Sapeornis* and *Fortunguavis* (IVPP V24192) plot with perching birds, and *Yanornis* plots with ground birds.

Discriminant predictions for LDA (Table 1) are consistent regardless of the inclusion of digit I, except for *Fortunguavis* and the Berlin *Archaeopteryx* specimen (MB.Av.101) where confidence decreases for the most likely prediction if digit I is excluded. *Ambopteryx*, *Confuciusornis*, and *Yanornis* are recovered as very likely to be specialised constricting raptors with some ground bird affinities. All three specimens of *Microraptor* are predicted as most likely having a restraining raptorial lifestyle, with striking and specialised constricting raptorial lifestyles the next most likely. One specimen of *Anchiornis* (STM 0-147) is recovered as equally likely to have a striking or specialised constricting raptorial lifestyle. The other two (STM 0-1, STM 0-114) are recovered as very likely to be ground birds. The Thermopolis *Archaeopteryx* (WDC-CSG-100) is recovered as most likely to be a striking or specialised constricting raptor. The Berlin *Archaeopteryx* (MB.Av.101) shows affinities with scavenging and ground birds. However, when digit I is excluded, restraining raptorial predation becomes the most likely result for MB.Av.101. *Sapeornis* is recovered as most likely to be a perching bird, with a lower likelihood of being either a constricting or striking raptor. *Fortunguavis* always shows some affinity with non-raptorial perching birds and restraint raptors. An affinity with striking raptors is only recovered when digit I is included. An affinity with specialised constricting raptors is only recovered when digit I is excluded. Discriminant analysis of principal components (DAPC) and its predictions are identical to LDA.

The exclusion of digit I from the data decreases the accuracy of the analysis, but precision is retained. In other words, the exclusion of digit I does not affect the ability to separate ecological categories, although modern birds are less likely to be correctly assigned. When using LDA equations to reclassify modern taxa, Fleiss' Kappa[47] comparing reclassifications to the true assignments was higher when digit I data were included (0.72 with digit I, 0.66 without). Phylogenetic honest significant differences (pairwise function comparing means) using the R package RRPP[48] recovered the same number of statistically different ecological groups regardless of the inclusion or exclusion of digit I (compare Table 2 with Table S1 in ref. 12). $K_{mult}$[49] for TM data is 0.60 with DI excluded, and 0.66 with DI included.

## Discussion

Historically, early theropod flyer ecology has been based on their anatomy, diet, aerial and terrestrial locomotion capabilities as well as the environments and climates they lived in[1–8].

Our analysis of their preserved pedal soft tissues and claws reveals a large diversity in foot morphology that, when compared to modern birds, allows us to constrain the ecological profiles of early theropod flyers as flight evolved. Starting with the early gliding-only scansoriopterygids[6], we find that the foot claws of *Ambopteryx* are weakly recurved with comparable size and superficial morphology to those of modern owls. However, *Ambopteryx* lacks opposed digits[41] suggesting it was not a grasping hunter[9]. Given this, the weakly recurved claws of *Ambopteryx* point to a ground-dwelling lifestyle. The pedal morphology of *Ambopteryx* is similar to the ground-dwelling oviraptorosaurians (Scansoriopterygidae has been recovered within Oviraptorosauria in some analyses[50–52]), so this morphology could be partly inherited from their common ancestor. The extreme climbing adaptations in the forelimbs of *Ambopteryx* and other scansoriopterygids[53] point to a scansorial ecological profile, so it may be that these were adequate for a scansorial lifestyle without specialisations in the hindlimb.

Non-avialan-powered flyers are represented by *Microraptor* in our dataset, which shows the most raptorial affinities within our sample. Its soft tissue (i.e., podotheca) morphology (Figs. 1 and 2), claw data using PCA and LDA (Fig. 7 and Table 1), preserved meals[54–56] and powered flapping flight potential[4] all suggest it was equipped to hunt flying and difficult-to-hold prey. The protrusive pads of *Microraptor* are located on both the tarsal pad and the proximal-most pad of digit IV. This is indicative of precise grasping capability, as protrusive pads provide a predatory advantage, acting like additional 'fingers' that penetrate the feathers/fur of struggling prey[10,57]. Strongly ginglymoid interphalangeal joints also serve to increase *Microraptor*'s grasping capabilities. These shared conditions with modern avivorous raptors suggest that their potential prey was birds, select non-avialan theropods, small pterosaurs, and gliding mammals[58]. The PCA and LDA analyses further indicate *Microraptor* was most likely a restraining raptor, using its enlarged digit II in a pinning role as in *Deinonychus*[34], consistent with past work on its long bones and non-claw phalanges[37]. However, digit I is problematic in this scenario as it is the smallest digit[59] and could not work in opposition to digit II to grip prey[9] (*contra*[60]). Instead, the main grip of digit II was probably assisted by *Microraptor* resting its body weight on the tarsometatarsus[34]. This hypothesis is supported by the small size of its preserved prey[54,55,61,62], which would be effectively pinned under *Microraptor*'s weight.

In *Anchiornis*, a ground-dwelling lifestyle and adaptations for taking non-volant prey are represented by podotheca soft tissue morphology (Figs. 1 and 3), claw data using PCA and LDA (Fig. 7 and Table 1), weakly ginglymoid interphalangeal joints (Fig. 1), preserved meals[63] and its relatively poor potential for powered flight[4]. 'Well-developed' toe pads in *Anchiornis* indicate raptorial capabilities associated with hunting ground-dwelling prey, as in modern birds[10]. The potential for weaker powered flapping flight in *Anchiornis* compared to other early flyers (including *Microraptor*[4]), supports a more ground-focussed lifestyle. This is consistent with weakly ginglymoid interphalangeal joints typical of cursorial avian and non-avian dinosaurs[34,36], and lizards and fish preserved in its digestive tract[63]. *Microraptor* and *Anchiornis* both retained an ancestral, functionally tridactyl foot with a reduced digit I. However, unlike *Microraptor*, *Anchiornis* lacks protrusional pads, suggesting that it was less capable of hunting volant prey, and which is further supported by its weaker flight capabilities[4]. The *Anchiornis* specimens show low claw curvature (*contra*[64]) and a small digit II claw relative to the digit III claw (also relative to the length of the non-claw phalanges[37]). Both of these characteristics do not indicate pinning/striking raptorial behaviour and are not found in modern scavenging vultures or perching birds. Specialised constricting

raptorial behaviour is mechanically unlikely as *Anchiornis* lacks fully opposed pedal digits (Fig. 3). Therefore, the podotheca data above as well as prior work indicating relatively weaker flight capability[4], both complement our PCA and LDA data in supporting a primarily terrestrial lifestyle for *Anchiornis*.

Although the podothecae are not preserved, the Berlin and Thermopolis specimens of *Archaeopteryx* have less recurved claws than most modern scavenging and perching birds. This is consistent with past work on claw curvature[30,65], non-ungual phalangeal proportions[66], limb bone length[67], and discrete locomotor traits[2] (*contra* past work on claw curvature[28,64]). This feature is similar to modern specialised constricting raptors or birds with a terrestrial lifestyle. Similarly sized claws, found only in the Thermopolis specimen, is another characteristic of modern specialised constricting raptors. However, the Thermopolis specimen is arguably more consistent with a terrestrial lifestyle as it lacks the 'enhanced' opposable digits (i.e., larger digit I or opposable digit IV) found in raptors that use their feet to grip prey[9]. The overall weakly ginglymoid interphalangeal and tarsometatarsal joints also indicate a pes that was poorly adapted for grasping. The digit III claw of the Berlin specimen is proportionally larger than most modern specialised constricting raptors or ground birds, consistent with non-claw phalangeal proportions[68]. Combined with the low curvature of all of its claws and primarily weakly ginglymoid interphalangeal articulations, this suggests an ecology with no modern analogue among our dataset. The Berlin and Thermopolis specimens have been proposed as separate species, particularly based on differences in claw morphology[43,69], which may explain their different ecological signals.

A generalist lifestyle was previously proposed for *Confuciusornis* using qualitative jaw observations and a fish consumulite[70], but the latter has been brought into doubt[1]. A generalist lifestyle is currently supported by different ecological signals from its claws, podotheca and jaws. Similarly sized, low-mid curvature claws indicate two possible lifestyles. Firstly, a specialised constricting raptorial lifestyle agreeable with past work on claw curvature[64] and non-claw phalanges[37]. Alternatively, a terrestrial lifestyle consistent with previous work on non-claw phalangeal proportions[66], limb bone length[67] and qualitative evaluation of the pes[71] (*contra* past work on limb bone length[72], claw curvature[65], discrete locomotor traits[2] and qualitative evaluation of the pes[73–76]) (Fig. 7 and Table 1). The few observable toe pads of *Confuciusornis* are low, indicating less raptorial affinities than *Microraptor* (Figs. 1, 2 and 4). Ginglymoid interphalangeal articulations in *Confuciusornis* point towards adaptations for grasping, though they do not distinguish between grasping for raptorial predation or non-predatory perching or climbing. Previous work shows that the jaw has a high mechanical advantage and stress profile similar to modern herbivores[8,77]. We also note that *Confuciusornis* was extremely common in its volcanically active, wet to semi-arid habitat[78]. Generalists tend to dominate volatile modern environments[79], further supporting our proposal of *Confuciusornis* as an ecological generalist.

Past work has shown *Sapeornis* to be an unexpectedly herbivorous[80–82] thermal soarer[80,83], as modern thermal soarers are mainly carnivorous[84]. This conflict between herbivory- and carnivory-related traits is also seen in our additional claw and toe pad data. *Sapeornis*' recurved claws and large digit III claw fits a non-raptorial perching bird, consistent with reconstructions of it as an arboreal herbivore. This is agreeable with past work on claw curvature[64,65], limb bone length[67], discrete locomotor traits[2], and qualitative evaluation of the pes[85]. 'Well-developed' toe pads (Figs. 1 and 5)[86] suggest a grasping capability, which could benefit carnivorous feeding. Highly ginglymoid interphalangeal joints also point to grasping capability, but as in *Confuciusornis*, do not necessarily indicate raptorial grasping. These unusual characteristics of *Sapeornis* suggest it was an ecologically complex herbivorous thermal soarer that supplemented its diet with meat, perhaps analogous to the modern palm-nut vulture (*Gypohierax*)[87].

*Fortunguavis* is an enantiornithine that was originally proposed as scansorial[45]. PCA and LDA claw data support a non-raptorial perching lifestyle. However, our dataset does not allow a climbing lifestyle to be tested as it does not distinguish climbing specialists from non-raptorial perching birds. In the PCA, the claws of *Fortunguavis* fall just outside those of non-raptorial perching birds (Fig. 7). In the original study[2,45] and our PCA, *Fortunguavis* plots near macaw parrots (*Ara*), which are birds that commonly scale trees and enclosure netting and were classified as aerial foragers rather than climbers[2]. Increasing ginglymoidy of more distal, non-claw phalanges also parallels the pattern in living macaws[88] and stem parrots[89], which we interpret as a compromise between flexibility and grip strength for arboreal locomotion. In LDA, *Fortunguavis* has a consistent affinity with perchers (Table 1).

Preserved soft tissues on the foot of *Yanornis* STM 9-531 are exposed in dorsal view (Fig. 6), which limited the inferences that could be made about this individual diet or behaviour. Although the lateral toe pad shapes are not observable, the pads are mesarthrally arranged on digit III, which is in contrast to all other taxa studied here. Mesarthral pads are found in modern non-raptorial birds[18,19], suggesting that *Yanornis* had a less efficient grasping capability than other early flyers in this study[10,27]. *Yanornis* IVPP V13558 has claws that are relatively straight, most similar to modern ground birds and specialised constricting raptors (Fig. 7 and Table 1). Specialised constricting raptors are noted as having relatively short phalanges compared to their claws[9], as opposed to *Yanornis* that has long phalanges and relatively short claws (compare[90,91]). A specialised constricting raptorial lifestyle is also inconsistent with the mesarthral toe pads and weakly ginglymoid interphalangeal joints in *Yanornis*. Thus, pedal data points towards a primarily ground-dwelling ecology, consistent with past work on avialan limb bone length[67] and reconstructions of Cretaceous ornithurans as non-arboreal[92] (*contra* work on non-claw phalanges[37]). Furthermore, preserved meals provide definitive evidence of *Yanornis* consuming fish[46,90], and are agreeable with its reconstruction as a non-arboreal bird.

Our study demonstrates that early theropod flyers had diverse ecological profiles. The more ground-dwelling profiles we recovered in the earliest Middle-Late Jurassic flyers *Ambopteryx, Anchiornis* and *Archaeopteryx* indicate that their weaker flight capabilities[3,4,6] did not permit a fully aerial lifestyle. *Ambopteryx* shows climbing and gliding forelimbs that were decoupled from its characteristically ground-adapted hindlimbs. We find that the Early Cretaceous non-avialan flyer *Microraptor* has the profile of a highly specialised aerial hunter. Current data shows this lifestyle was unique within the Jehol Biota, and among all other known early theropod flyers. This suggests that specialised predatory roles filled by birds in modern ecosystems[93] were performed by non-bird flyers like *Microraptor*. *Confuciusornis* is recovered as a generalist, while *Fortunguavis* is recovered as a perching bird, and *Yanornis* as primarily terrestrial. Other early flyers have ecological profiles different from any modern birds in this study that would be valuable to explore further, e.g., Berlin *Archaeopteryx* and *Sapeornis*. Generalists typically survive over specialists in times of ecosystem crisis[94]. Our results reveal early specialist non-avialan and avialan flyers that would be more susceptible to extinction during such crises. This should be taken into account as we work to better understand the turnover of theropod flyers and the rise of modern birds.

## Methods

### Institutional abbreviations
CMNH, Carnegie Museum of Natural History, Pittsburgh, United States; FMNH, Florida Museum of Natural History, Gainesville, United States; HGM, Henan Geological Museum, Zhengzhou, China; IVPP, Institute of Vertebrate Paleontology & Paleoanthroplogy, Beijing, China; MB, Museum für Naturkunde Berlin, Germany; STM, Shandong Tianyu Museum of Nature, Pingyi, China; WDC, Wyoming Dinosaur Center, Thermopolis, United States.

### Permits and ethics approval
No relevant permits were needed for the work in this study. The collections visited made specimens available for study according to their own ethical guidelines. These specimens were studied according to the ethical guidelines of these collections.

### Fossil specimen selection
Over 1000 early paravian fossils from the Shandong Tianyu Museum of Nature were imaged using LSF[40] in search of preserved pedal soft tissues (see 'Laser-stimulated fluorescence' section for more details). This revealed 12 specimens with exceptionally preserved foot pads that were selected and analysed. This involved studying the arrangement and proportions of the toe pads and scales of the podotheca, and the proportions and geometry of the pedal phalanges and claws. Previously published specimens of *Confuciusornis*[38] and *Sapeornis*[95] with preserved foot pads were also incorporated into the study, and claw measurements from an additional specimen of *Yanornis* (IVPP V13558) were sourced from the literature[46] as our focal *Yanornis* pes (STM 9-531) is preserved in an orientation where claw morphometry was not possible. While no known scansoriopterygid or enantiornithine specimens preserve portions of the podotheca, we included claw measurements from *Ambopteryx*[41] and *Fortunguavis*[45] in order to more broadly comment on the ecology of theropod flight evolution. For the same reason, we also included photos of the Berlin and Thermopolis specimens of *Archaeopteryx* taken by M.P. and T.G.K.

### Modern specimen selection
Podothecae of 15 modern bird of prey species (spanning Pandionidae, Accipitridae, Tytonidae, Strigidae and Falconidae) and an additional sample of 21 species of modern non-predatory birds (spanning Psittaciformes, Passeriformes, Caprimulgiformes and Coraciiformes) were studied in the collections of the University of New England Natural History Museum, Armidale, Australia and the Ornithology Collection of the Australian Museum Research Institute, Sydney, Australia. Measurements and photographs were collected in person at the University of New England by P.R.B. and N.J.E., and at the Australian Museum by L.R.T. See Tsang et al.[10] and Supplementary Information herein for additional details.

Claws from 61 taxa covering a wide variety of crown bird families (particularly from raptorial birds) were sampled from the skeletal collections of the Carnegie Museum of Natural History, Pittsburgh, United States and the Florida Museum of Natural History, Gainesville, United States. See ref. 12 and Supplementary Information herein for additional details. All claws were checked for high porosity, which is a common pathology among captive birds in enclosures with hard flooring (see also ref. 96). Measurements and photographs were collected in person by C.V.M.

### Laser-stimulated fluorescence (LSF)
Fossil specimens were imaged under LSF to reveal additional soft tissue details that were not visible under white light conditions, following the methodology of Wang et al.[15] based on Kaye et al.[40]. A 405 nm violet near-UV laser diode was used to fluoresce the specimen according to standard laser safety protocol. Long exposure photographs were taken in a darkened room with a Nikon D810 DSLR camera fitted with a 425 nm blocking filter. Image post-processing (equalisation, saturation and colour balance) was performed uniformly across the entire field of view in Photoshop CS6.

### Integumentary terminology
Integumentary terminology for the podotheca (including scutate, scutellate, and reticulate scales) follows Lucas and Stettenheim[16]. Terminology for features on the plantar surface of the pes follows Lennerstedt[18,19] and Tsang et al.[10]. Toe pads can be flat (low profile; flat plantar pad surface in lateral aspect), well-developed (high profile;

convex plantar pad surface in lateral aspect with a semi-lenticular outline) or protrusional (very high profile; highly convex plantar pad surface with a semi-oval or semi-circular outline in lateral aspect; forming 'fingers') (Supplementary Fig. 1). A furrow forms the 'hinge' between toe pads, or between a toe pad and a fold. Furrows can be narrow and V-shaped or broad with an obvious flattened surface between successive toe pads. Folds are small, raised areas that occur between larger toe pads, and as a result, do not extend farther ventrally than the main toe pads (Supplementary Fig. 1).

### Traditional morphometrics (TM)
**Ecological category assignment.** Ecological categories for raptorial birds and their assignment generally follow[9], with elaboration from[12]: Restraint - hawks, eagles (Accipitridae), forest-falcons (*Micrastur)*, shrikes (Laniidae), and helmetshrikes (Vangidae) use their talons for prolonged prey restraint while they kill large prey slowly; Strike – falcons (Falconini), secretarybirds (*Sagittarius*) and seriemas (Cariamidae) are all known to use their feet for high-speed concussive strikes; Constriction - owls (Strigidae) are specialised to constrict small animals within their toes, mainly using talons to extend their reach; Pierce – ospreys (*Pandion*) pierce their talons into fish to aid in gripping as they extract them from the water. Our Constrict category supersedes the 'Suffocate' category in[12]. We renamed this category because constriction is a more encompassing term to use. This is because while constriction leading to suffocation has been confirmed in some birds[97,98] some constrictors outside of birds have been shown to kill prey by cutting off their blood flow[99].

Among non-raptorial birds, birds in the 'ground' and 'perching' categories spend the vast majority of their time on the ground or perched on a branch, respectively. Clades included in these groups follow[12], and were selected to provide wide phylogenetic breadth.

**Measurements.** Measurements and landmarks for TM follow ref. 9 with modifications from ref. 12 that permits their application to more fossil taxa. The seven parameters used in the TM analysis were outer arc curvature (Oo in °) for each digit (I, II, III and IV) and outer arc length (ALo) of digits I, II and IV expressed as a unitless ratio to the outer arc length of digit III (*sensu*[9]). Most fossil specimens in this study do not preserve digit I, so separate datasets were created both including and excluding this digit.

Linear measurements of modern claws were taken with a tape measure to avoid damaging the bones, except for measurements less than 1 cm which were collected with callipers. Photos were taken in lateral view and imported into CorelDraw X8 for angular measures using the 'Angular Dimension' tool. Photos taken at different angles were used to test for parallax effects, with angular measures not notably affected. In cases where digit identifications were uncertain, the claws were compared to taxidermy specimens from the same collection and referred to a digit based on their relative size and curvature.

**Tree topologies.** Modern portions of the phylogenetic tree used in this study for phylogenetic mapping and honest significant difference correction were taken from birdtree.org[100]. Non-avian paravian branches were then grafted onto the modern tree following the topology of ref. 50. All species were placed at the age of their oldest discovery, with species divergences arbitrarily taking 10,000 years. All grafted lengths were scaled so that the total length of the modern portion of the tree was equal to 94 Ma based on an estimated origin of Aves by ref. 101.

**Data analysis.** As several fossil specimens do not preserve digit I, two separate datasets are needed so that all specimens can be used: one dataset that includes digit I measurements and another that excludes them. Two primary analyses were performed on each of these morphometric datasets: principal component analysis (PCA) and linear discriminant analysis (LDA). Both analyses reduce the dimensionality of

the data making it easier to interpret, but do so with different goals. PCA maximizes the variance explained by the axes while LDA maximizes the separation of a priori groups[102] (in this case ecological categories). All PCAs in this study use the correlation matrix, which removes the effects of units and scale by scaling inputs to constant variance.

Several adjustments need to be made for using LDA in this context. LDA requires all data to be uncorrelated, which is unlikely in biological data. We follow[12] and use discriminate analysis of principal components (DAPC)[103] to account for this. The LDA and DAPC results are identical showing that LDA is robust to the uncorrelated assumption. In addition, *Pandion haliaetus* was excluded from all LDA and phylogenetic honest significant differences as neither is designed to analyse groups with only one member.

TM variables of modern groups with more than one member were compared for significant differences using the pairwise() function (phylogenetic honest significant differences *sensu*[12]) in the R package RRPP[48] version 1.1.2 in R version 4.1.2 (Table 2). A total of 1000 permutations were used by convention, with sensitivity analyses finding $p$ values to converge before this point. $K_{mult}$, a statistic summarising the phylogenetic signal in multivariate data, was calculated using the R code in ref. 49.

### Fossil interpretation
All of the fossil paravians used in this study are exposed on the bedding plane and visible in a single, roughly two-dimensional orientation. Distortion of the bones is limited to crushing as a result of diagenetic compression of the entombing sediments. In some specimens (e.g., STM 0-7, STM 1-114, STM 5-109), minor disarticulation of the phalanges attests to a degree of pre-burial decay and rupture of the soft tissue envelope surrounding the bones. In each of these cases, disarticulation is limited to a single non-claw phalanx. In other cases (e.g., STM 0-114, STM 0-147), minor or major differences in the orientation of the digits do not permit a perfect lateral view of the toe pads and podotheca, which is the preferred orientation for determining toe pad protrusion. Importantly, lateral expansion and/or deformation of the soft tissue outlines due to compression are not considered the norm in fossil samples[104,105]. More specifically, decay, compression, differences in orientation, or any combination of these, and other taphonomic factors, are not considered significant biases, as the observed soft tissue outlines are frequently replicated (1) between left and right pedes, and/ or (2) between specimens. Where these effects were more obvious (especially in the case of digit orientation) and where soft tissue outlines were ambiguous (e.g., in regions where two or more digits partially overlap), these digits/toe pads were not analysed herein.

### Reporting summary
Further information on research design is available in the Nature Portfolio Reporting Summary linked to this article.

## Data availability
The images and all other data pertinent to this research are available in the main text and Supplementary Information. Source Data are provided with this paper. These data can also be obtained from the corresponding authors. The fossil specimens investigated are available for scientific study by qualified researchers at the Carnegie Museum of Natural History (Pittsburgh, United States), Florida Museum of Natural History (Gainesville, United States), Henan Geological Museum (Zhengzhou, China), Institute of Vertebrate Paleontology & Paleoanthroplogy (Beijing, China), Museum für Naturkunde Berlin (Berlin, Germany), Shandong Tianyu Museum of Nature (Pingyi China) and the Wyoming Dinosaur Center (Thermopolis, United States). Supplementary Data 1 comprises of modern avian toe pad and foot scale data as well as traditional morphometric claw data for modern birds and early theropod flyers.

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

## Acknowledgements
This study was supported by the General Research Fund of the Research Grant Council of Hong Kong (RGC GRF 17103315, 17120920 and 17105221; M.P.), the School of Life Sciences of The Chinese University of Hong Kong, a Postgraduate Scholarship from The University of Hong Kong (PGS; C.V.M.), a Research Training Program scholarship from the Australian Government (RTP; N.J.E.), the Taishan Scholars Program of Shandong Province (Ts20190954; X.L.W.) and the National Natural Science Foundation of China (42288201; X.T.Z.). T. Alexander Dececchi is thanked for the discussions about this manuscript.

## Author contributions
M.P., X.L.W. and T.G.K. initially designed the project. M.P. and T.G.K. collected the main project data (i.e., performed specimen imaging) with input from X.L.W., X.T.Z., P.R.B., M.L., C.V.M., N.J.E., L.R.T. and Y.T.T. All authors assessed and analysed the data. M.P., P.R.B., C.V.M., N.J.E., Y.T.T. and T.G.K wrote the paper with input from X.L.W., X.T.Z., L.R.T. and M.L.

## Competing interests
The authors declare no competing interests.
