## [Peer Review File · Nature Communications]

Exceptional Preservation and Foot Structure Reveal Ecological Transitions and Lifestyles of Early Theropod FlyersReviewers' Comments:

Reviewer #1:

Remarks to the Author:

An important study that fills an obvious gap. Feet have been neglected for too long. The study demonstrates the huge value of feet as paleobiological data source, and there are certainly more parameters in foot skeletons that could be explored in future studies. Making the connection to the fossil track record might be one of the next steps, too.

My comments below:

- I strongly encourage the authors to make data publicly available. If images cannot be made available, this is unfortunate but I understand there may be political reasons. But at the very least, I suggest to make the data available that was used to perform the statistical analysis. This is simply necessary to allow others to reproduce and validate the results.
- In the method section, we all of a sudden hear about "interdigital variation", but I don't remember that this was introduced in the introduction, and I do not see it explained elsewhere. What may this parameter tell us? What do modern birds tell us about this?
- From the record of tridactyl dinosaur tracks, it appears that pad counts are very conservative (usually 2-3-4 in digits II, III, and IV, respectively), but much more variable in birds. Would be interesting to see how variable pad counts vary in the forms you analyse?
- Line 85: You imply a clear-cut distinction between arthral and mesarthral in birds, but in birds, pads are quite variable, and often do not correlate to phalanges precisely (one pad may incorporate several phalanges in cases), see the works of Jim Farlow.
- Line 152: Which Archaeopteryx specimen did you study in China? Only the Berlin and Thermopolis specimen are mentioned elsewhere in the manuscript.
- Figure order is seemingly off: Confuciusornis is only Fig. 6 even though it is discussed Sapeornis in the text
- Line 315: In which figure or table can we find the LDA results? Table 1? Please refer to it.
- Line 342: "a ground birds" - typo
- Line 371-372: Oviraptorosaurians were certainly not the most closely related taxon to Ambopteryx, and I even doubt that scansoriopterygids as a whole were, since oviraptorosaurians are more basal?
- line 386 - this is repetitive, was mentioned before.

Jens N. Lallensack

Reviewer #2:

Remarks to the Author:

I had favorably reviewed a previous version of this manuscript a few months ago, suggesting "accept with minor revisions". However, at the time, I did make some suggestions that the authors might perhaps elucidate further evidence for their ecological interpretations by studying the degree of ginglymoidy on the phalangeal and metatarsal articulations (and some other physical features). I also made some minor suggestions on some specific terminology.

In this new version of the manuscript, the terminology change has been implemented. Further, the authors have now included details of ginglymoidy where it is recordable. This is a good addition to the paper and strengthens certain ecological interpretations - or at least adds a new dimension for future analysis.

Most of the review that follows is thus for the benefit of the editor, and is mostly copied and pasted from my previous review, as the basic structure of the paper is the same as before. I have added in a few new parts, but nothing major.

Just to clarify – the study which I am being asked to review is a mostly qualitative analysis of ~12 specimens (not all are figured) of basal paravians which show well-preserved podotheca and other scales of the feet and hindlimb. This is combined with data on foot claw size and curvature, and degree of ginglymoidy of phalanx/metatarsal articulations to infer ecology. It is my understanding that the claw data is derived from a larger study (Miller et al.) which is not yet published, but is presumably in review/press etc., and which was provided as supplemental information for the purposes of this review.

The authors utilize variable light wavelengths to reveal striking images of the feet, showing scales and podotheca that are either invisible or poorly visible to the naked eye. The authors describe the extent of the podotheca, types of scales, positions and size of pads and furrows, published functional morphology and diet data, measure the curvature and size of the claws, and report the degree of ginglymoidy of articular ends of the pedal phalanges and metatarsals. These observations and data are compared with extant birds to infer ecology of the extinct species.

The article is well written. I have no problem with the language and grammar. Overall I really like the paper and it should be published. I am not a statistician, so I do not comment on that aspect.

The research is commendably innovative. The fabulous figures of the feet are certainly worthy of publication in their own right, but the authors take a step beyond the mere spectacle of the images and rightly see the fossilized podotheca as an opportunity to look at these species in a way that has not been done before. There is thus much to like about the authors approach.

METHODS - D-III ONLY VS USING ALL TOES

It was nice to see that the authors did not just rely on D-III claw curvature. I have reviewed so many manuscripts that measure only D-III, and it has been frustrating, since D-III is the worst single digit to use. To be fair, measuring D-III alone is what most authors did in the 1990's-2000's, but this is really not much of an excuse for current authors not to be thinking about this choice. I partly blame myself for not stating this clearly enough in our own papers.

I recall reading in the Miller review paper that Hedrick et al (2019) considered D-III to be usable alone. Well, it is "usable" but it is not optimal. Digit III bears the most bodyweight when walking and it's on the midline, so it is almost always a "walking" toe, and has proportions that reflect this (even in perching birds) – the claw is also the least curved (generally). What is quite amazing to me is that some authors have even specifically noted this, but continue to use D-III anyway, without seeming to realize or consider how it would affect their data. When undergoing the tedious process of measuring hundreds or thousands of specimens, I wonder if they ever looked at the other toes of the specimens they are measuring and notice perhaps how different D-II and D-IV look among taxa, and wonder why this might be so? D-II and D-IV are "accessory" digits. If the bird is doing anything interesting behaviourally with its feet, it should show in D-II or IV, much more strongly than III. Behaviours that occur on the centre (under the head) are reflected in D-II. Behaviours that require lateral interaction (e.g. stabilizing when moving, increase in grasping area) are reflected in differences in D-IV. D-III is going to vary the least among taxa, so if it is used alone then resultant groups are going to be less distinct (and more subject to slight measurement inaccuracies since there is a narrower spread of angle measurement overall). I expect broad data trends can be ascertained from D-III curvature (ie. Hedrick's et al.'s paper), but to get more detailed indicators (ie ecological categories) you need relative differences with D-II and IV (and I, although this is a more specialized digit in many ways).

So like I say, I commend the authors for looking at all digits. They are, I think, only the second set of people to do so in papers I have read or reviewed since we published in 2011.

METHODS – OTHER

As I said before, the authors have added new data in the form of degree of ginglymoidy to accompany what might be considered the more traditional method of curvature. In a previous review, I criticized authors of previously published papers in using only curvature data. So again, here the current authors are ahead of the curve, and I commend them for this.

RESULTS

The ecological inferences are now more robust, being based on more data. I still expect there to be future amendments to these ecological interpretations, as more functional characters are recognized and assessed -this is still quite an open subject, even for living taxa.

A few line edits:

LINE 342: very likely to be a ground birds

Delete "a"

LINE 193-195: Distal facets of metatarsal II and metatarsal III in STM 5-109 appear to range from weakly to strongly ginglymoid

Which is which? Can you be more specific? Is D-II strongly ginglymoid and D-III not?

LINE 114-115: Constricting raptors tend to have slightly increased claw curvature and claws of subequal size9,12,30

Owls (specialised constrictors) do yes, but constriction is also used by hawks and falcons when immobilising prey that can be held within the foot (Fowler et al., 2009). You could maybe make an argument that specialised constriction may tend towards these proportions.

SUMMARY

I think this paper (and Miller's broader work) has two aspects – 1. awesome imagery from a clever methodology facilitating a genuinely unique new avenue of investigation (podotheca), combined with 2. a step forward in our understanding of interpreting theropod foot functional morphology. However, (2) is largely a result of coming after some significant backward steps in recent research on claws – mainly the reduction in studying multiple ecological indicators, towards an overfocus on curvature and only D-III. Perhaps (2) is better stated as this manuscript helps get theropod foot morphology studies back on track.

I think that the study would be of immediate interest to people who work on dinosaurs, and given the striking images and cleverness of the approach, it would be of interest to a broader group of scientists and the public, so it is appropriate for this journal.

I wish to reveal my identity to the authors: Denver Fowler

Reviewer #3:

Remarks to the Author:

Comments to the authors

I have now considered the manuscript entitled "Feet of Early Theropod Flyers Show Broad Ecological Transitions from Ground-Based Through Aerial Lifestyles". This paper investigates the ecology of

earliest theropod fliers, based on the pedal soft tissues and claws. The method consists in a morphological analysis including modern birds to infer the fossil ecology.

Considering the relative lack of information about early theropod locomotion, the paper tackles a relevant and interesting topic. The fossil dataset seems relevant to address this type of question, as it includes a diversity of potential ecology (based on the previous literature), a diversity of morphology and a relatively high number of specimens. A description of toe pads, foot scales, claws and joints is presented for the fossil sample and compared to modern birds, with the associated implication on fossil ecology.

The manuscript could be streamlined to improve readability. Lots of results are presented, especially anatomical descriptions, which is a strength. Yet, the authors do not clearly mention the impacts of their findings in the abstract and introduction. Unfortunately, it gives the impression of a list of results which are not enough put in perspective with previous works.

The structure of the introduction should be improved. As it is, the author's contribution is difficult to find and the knowledge gap is not obvious, especially because the fossils were previously described and the material previously analysed. The paragraph listing all the potential anatomical adaptations related to the locomotor abilities is long and somewhat related to the material and methods and / or discussion sections. Some important references are missing, especially on the curvature of the hallux claw, related to the climbing or perching ability.

The beginning of the discussion section resembles of a list of findings based on the list of fossils observed. The paper would be strengthened by a clear question addressed and answered in the paper. The method is not detailed. I understand it is extensively described in a manuscript under review in another journal, but there could be a summarized section in the present manuscript to briefly explain it. It would strengthen the results and the conclusions.

Similarly, the current manuscript does not describe the modern bird sample enough. As written, it is difficult to immediately see the phylogenetic diversity, number of species and number of modern bird individuals used in the study. A table should be added. In addition, it lacks a section describing the limits of the study (problem with assessing a discrete ecology category to an animal, problem with the lack of certain bones, problem with position in the fossil, etc.), which would mitigate the impact of the results.

Please, take note of the chronological comments and suggestions below.

Title

The title does not provide a clear view of the main result. As it is, it is too broad.

Abstract

The abstract does not provide an accessible summary of the paper. There is no clear mention of the methods. The end part is a list of fossils with potential inferred ecologies, but the results and impact of the present study is not clearly stated.

Line 43 page 2: "We interpret these foot data in the context of existing lines of evidence to helps us better understand the evolutionary ecology of early theropod flyers."

Not informative sentence

Line 46 page 2: "more". Not clear to what it is compared, please rephrase.

Introduction

Line 63 page 3 : "by comparing their toe pads, foot scales, claws and joints with living birds"
How do you compare? What methods? Please clarify

Line 69 page 4: "In modern birds, morphological differences in the scales and toe pads have been correlated to locomotory and feeding preferences"

How do they correlate? What feature is used and in what direction is the morphological is going?

Line 107 page 5: "The morphometric dataset used here¹² was collected using the same measurements as 9,30."

Please briefly explain what the method is.

Line 108 page 5: "This second dataset has a broader overall phylogenetic coverage, applies ecological categories to a broader range of taxa and uses bone-based landmarks that better apply to fossils"

Not clear what "this second dataset" is referring to. Are you talking about your dataset?

Please provide exact numbers to show how the dataset was improved

Line 116 page 6: "Non-raptorial perching birds have strongly curved claws¹²"

Not all non-raptorial perching birds do, in Dendrocolaptinae and Certhiidae the claw of the hallux is relatively straight while they climb upward on vertical surfaces.

Bock, W. J. and Miller, W. D. (1959). The scansorial foot of the woodpeckers, with comments on the evolution of perching and climbing feet in birds. *American Museum novitates*; no. 1931.

A reference about the different methods for assessing the shape of the claw should be added too: Tinius, A., & Patrick Russell, A. (2017). Points on the curve: an analysis of methods for assessing the shape of vertebrate claws. *Journal of morphology*, 278(2), 150-169.

As well as a reference on the functional morphology of vertebrate claws.

Thomson, T. J. and Motani, R. (2021). Functional morphology of vertebrate claws investigated using functionally based categories and multiple morphological metrics. *Journal of Morphology Wiley* 282, 449–471.

Results

Line 151 page 7: "This collection revealed 12 specimens with exceptionally preserved toe pads, foot scales and claws belonging to the early flyers Anchiornis, Archaeopteryx, Confuciusornis, Microraptor, Sapeornis, and Yanornis"

What were the criteria to choose these specimens? Was phylogenetic diversity a criteria? The authors mentioned the good preservation, but later in the manuscript, they say: Line 165 page 8: "However, this observation is uncertain due to the poor preservation of the phalanges."

Please clarify.

Discussion

Line 481 page 22: This paragraph should come earlier in the Discussion section, as it corresponds to the description of the overall story formed.

In general, the discussion should be revised to put forward the main results and contribution.

Images, Graphs and Data Tables

Figure 1. This figure should be improved. The legends corresponding to "arthrally" and "mesarthral" are too close to each other to clearly show the difference between the two. As it is something mentioned in the paper, this should be easily identified in the figure.

Figure 4 lacks a picture of the specimen.

A table with the dataset used for modern birds is missing (could be in supplementary materials)

Material and Methods

Line 709 page 40: "See 12 for additional details".

A list of the specimens with the name, family, collection number would be useful.

Line 715 page 40: "the methodology of Wang et al. 96 based on Kaye et al. 36."

Same remark, having a sentence to explain the method would be useful.

References

There are over 100 references, but important ones are missing. Please reconsider your list to make it more relevant to the paper.

RESPONSE TO REVIEWER COMMENTS

> We thank the reviewers for their time in reviewing our work and for the comments they provided. We used this feedback to make a wide range of manuscript edits that we believe have improved the quality of our manuscript. Please find our point-by-point responses below.

Reviewer #1 (Remarks to the Author):

An important study that fills an obvious gap. Feet have been neglected for too long. The study demonstrates the huge value of feet as paleobiological data source, and there are certainly more parameters in foot skeletons that could be explored in future studies. Making the connection to the fossil track record might be one of the next steps, too.

My comments below:

– I strongly encourage the authors to make data publicly available. If images cannot be made available, this is unfortunate but I understand there may be political reasons. But at the very least, I suggest to make the data available that was used to perform the statistical analysis. This is simply necessary to allow others to reproduce and validate the results.

> This is a very important point and we agree. We have included specimen images needed to reproduce our observations and validate our results in the main figures of the paper. We have added the traditional morphometrics dataset into Supplementary File 1, which was omitted accidentally in the original submission.

– In the method section, we all of a sudden hear about "interdigital variation", but I don't remember that this was introduced in the introduction, and I do not see it explained elsewhere. What may this parameter tell us? What do modern birds tell us about this?

> We have changed the wording to explicitly convey our original intended meaning. Thanks for bringing this up.

– From the record of tridactyl dinosaur tracks, it appears that pad counts are very conservative (usually 2-3-4 in digits II, III, and IV, respectively), but much more variable in birds. Would be interesting to see how variable pad counts vary in the forms you analyse?

> This is an interesting point that we will be keen to explore in a future paper.

– Line 85: You imply a clear-cut distinction between arthral and mesarthral in birds, but in birds, pads are quite variable, and often do not correlate to phalanges precisely (one pad may incorporate several phalanges in cases), see the works of Jim Farlow.

> Yes, we agree that there is nuance that should be better conveyed. To this end, we have added more information to the relevant paragraph, including citations to the works of Farlow.

– Line 152: Which Archaeopteryx specimen did you study in China? Only the Berlin and Thermopolis specimen are mentioned elsewhere in the manuscript.

> Thanks for spotting this. This was a typo and has been deleted.

– Figure order is seemingly off: Confuciusornis is only Fig. 6 even though it is discussed Sapeornis in the text

> Thanks for spotting this. The text order did not line up with the figure label order. This has been corrected.

– Line 315: In which figure or table can we find the LDA results? Table 1? Please refer to it.

> Thanks for spotting this. We have edited the text accordingly.

– Line 342: "a ground birds" – typo

> Thanks for spotting this. We have corrected this.

– Line 371–372: Oviraptorosaurians were certainly not the most closely related taxon to Ambopteryx, and I even doubt that scansoriopterygids as a whole were, since oviraptorosaurians are more basal?

> Thanks for this comment. We believe that scansoriopterygids are sister to oviraptorosaurians, but note that this is not yet a consensus view (See discussion in Pittman et al 2022: 440-01-pittman_et_al.pdf (amnh.org)). We have edited the sentence appropriately.

– line 386 – this is repetitive, was mentioned before.

> Thanks for spotting this. We agree and have deleted the repetition on this line.

Jens N. Lallensack

Reviewer #2 (Remarks to the Author):

I had favorably reviewed a previous version of this manuscript a few months ago, suggesting “accept with minor revisions”. However, at the time, I did make some suggestions that the authors might perhaps elucidate further evidence for their ecological interpretations by studying the degree of ginglymoidy on the phalangeal and metatarsal articulations (and some other physical features). I also made some minor suggestions on some specific terminology.

In this new version of the manuscript, the terminology change has been implemented. Further, the authors have now included details of ginglymoidy where it is recordable. This is a good addition to the paper and strengthens certain ecological interpretations – or at least adds a new dimension for future analysis.

Most of the review that follows is thus for the benefit of the editor, and is mostly copied and pasted from my previous review, as the basic structure of the paper is the same as before. I have added in a few new parts, but nothing major.

Just to clarify – the study which I am being asked to review is a mostly qualitative analysis of ~12 specimens (not all are figured) of basal paravians which show well-preserved podotheca and other scales of the feet and hindlimb. This is combined with data on foot claw size and curvature, and degree of ginglymoidy of phalanx/metatarsal articulations to infer ecology. It is my understanding that the claw data is derived from a larger study (Miller et al.) which is not yet published, but is presumably in review/press etc., and which was provided as supplemental information for the purposes of this review.

The authors utilize variable light wavelengths to reveal striking images of the feet, showing scales and podotheca that are either invisible or poorly visible to the naked eye. The authors describe the extent of the podotheca, types of scales, positions and size of pads and furrows, published functional morphology and diet data, measure the curvature and size of the claws, and report the degree of ginglymoidy of articular ends of the pedal phalanges and metatarsals. These observations and data are compared with extant birds to infer ecology of the extinct species.

The article is well written. I have no problem with the language and grammar. Overall I really like the paper and it should be published. I am not a statistician, so I do not comment on that aspect.

The research is commendably innovative. The fabulous figures of the feet are certainly worthy of publication in their own right, but the authors take a step beyond the mere spectacle of the images and rightly see the fossilized podotheca as an opportunity to look at these species in a way that has not been done before. There is thus much to like about the authors approach.

> Thanks very much for mentioning your previous favourable review and for acknowledging the changes suggested have been implemented in this manuscript version.

METHODS - D-III ONLY VS USING ALL TOES

It was nice to see that the authors did not just rely on D-III claw curvature. I have reviewed so many manuscripts that measure only D-III, and it has been frustrating, since D-III is the worst single digit to use. To be fair, measuring D-III alone is what most authors did in the 1990's-2000's, but this is really not much of an excuse for current authors not to be thinking about this choice. I partly blame myself for not stating this clearly enough in our own papers.

I recall reading in the Miller review paper that Hedrick et al (2019) considered D-III to be usable alone. Well, it is “usable” but it is not optimal. Digit III bears the most bodyweight when walking and it's on the midline, so it is almost always a “walking” toe, and has proportions that reflect this (even in perching birds) – the claw is also the least curved (generally). What is quite amazing to me is that some authors have even specifically noted this, but continue to use D-III anyway, without seeming to realize or consider how it would affect their data. When undergoing the tedious process of measuring hundreds or thousands of specimens, I wonder if they ever looked at the other toes of the specimens they are measuring and notice perhaps how different D-II and D-IV look among taxa, and wonder why this might be so? D-II and D-IV are “accessory” digits. If the bird is doing anything interesting behaviourally with its feet, it should show in D-II or IV, much more strongly than III. Behaviours that occur on the centre (under the head) are reflected in D-II. Behaviours that require lateral interaction (e.g. stabilizing when moving, increase in grasping area) are reflected in differences in D-IV. D-III is going to vary the least among taxa, so if it is used alone

then resultant groups are going to be less distinct (and more subject to slight measurement inaccuracies since there is a narrower spread of angle measurement overall). I expect broad data trends can be ascertained from D-III curvature (ie. Hedrick's et al.'s paper), but to get more detailed indicators (ie ecological categories) you need relative differences with D-II and IV (and I, although this is a more specialized digit in many ways).

So like I say, I commend the authors for looking at all digits. They are, I think, only the second set of people to do so in papers I have read or reviewed since we published in 2011.

> Thanks very much. Your comments show that our efforts were really worthwhile.

METHODS – OTHER

As I said before, the authors have added new data in the form of degree of ginglymoidy to accompany what might be considered the more traditional method of curvature. In a previous review, I criticized authors of previously published papers in using only curvature data. So again, here the current authors are ahead of the curve, and I commend them for this.

> Thanks very much.

RESULTS

The ecological inferences are now more robust, being based on more data. I still expect there to be future amendments to these ecological interpretations, as more functional characters are recognized and assessed -this is still quite an open subject, even for living taxa.

> Thanks for acknowledging that the changes you recommended for a previous version of the manuscript have been implemented in the current manuscript version.

A few line edits:

LINE 342: very likely to be a ground birds

Delete "a"

> Thanks for spotting this. We have corrected this.

LINE 193-195: Distal facets of metatarsal II and metatarsal III in STM 5-109 appear to range from weakly to strongly ginglymoid

Which is which? Can you be more specific? Is D-II strongly ginglymoid and D-III not?

> Thanks for spotting this. We have corrected this.

LINE 114-115: Constricting raptors tend to have slightly increased claw curvature and claws of subequal size^{9,12,30}

Owls (specialised constrictors) do yes, but constriction is also used by hawks and falcons when immobilising prey that can be held within the foot (Fowler et al., 2009). You could maybe make an argument that specialised constriction may tend towards these proportions.

> Great point. We have therefore gone through the MS and changed instances of “constricting raptors” to “specialised constricting raptors”. We have also been more specific about our usage in the details added in brackets.

SUMMARY

I think this paper (and Miller’s broader work) has two aspects – 1. awesome imagery from a clever methodology facilitating a genuinely unique new avenue of investigation (podotheca), combined with 2. a step forward in our understanding of interpreting theropod foot functional morphology. However, (2) is largely a result of coming after some significant backward steps in recent research on claws – mainly the reduction in studying multiple ecological indicators, towards an overfocus on curvature and only D-III. Perhaps (2) is better stated as this manuscript helps get theropod foot morphology studies back on track.

I think that the study would be of immediate interest to people who work on dinosaurs, and given the striking images and cleverness of the approach, it would be of interest to a broader group of scientists and the public, so it is appropriate for this journal.

I wish to reveal my identity to the authors: Denver Fowler

Thank you very much Denver for the wonderful comments and for acknowledging the changes we made to this manuscript based on a previous favourable review you gave for a different journal.

Reviewer #3 (Remarks to the Author):

Comments to the authors

I have now considered the manuscript entitled “Feet of Early Theropod Flyers Show Broad Ecological Transitions from Ground-Based Through Aerial Lifestyles”. This paper investigates the ecology of earliest theropod fliers, based on the pedal soft tissues and claws. The method consists in a morphological analysis including modern birds to infer the fossil ecology.

Considering the relative lack of information about early theropod locomotion, the paper tackles a relevant and interesting topic. The fossil dataset seems relevant to address this type of question, as it includes a diversity of potential ecology (based on the previous literature), a diversity of morphology and a relatively high number of specimens. A description of toe pads, foot scales, claws and joints is presented for the fossil sample and compared to modern birds, with the associated implication on fossil ecology.

The manuscript could be streamlined to improve readability. Lots of results are presented, especially anatomical descriptions, which is a strength. Yet, the authors do not clearly mention the impacts of their findings in the abstract and introduction. Unfortunately, it gives the impression of a list of results which are not enough put in perspective with previous works.

> We have further improved the readability and made the changes suggested.

The structure of the introduction should be improved. As it is, the author’s contribution is difficult to

find and the knowledge gap is not obvious, especially because the fossils were previously described and the material previously analysed. The paragraph listing all the potential anatomical adaptations related to the locomotor abilities is long and somewhat related to the material and methods and / or discussion sections. Some important references are missing, especially on the curvature of the hallux claw, related to the climbing or perching ability.

> We made improvements based on the comments given.

The beginning of the discussion section resembles of a list of findings based on the list of fossils observed. The paper would be strengthened by a clear question addressed and answered in the paper.

> We have made further improvements to the discussion and the narrative of the paper.

The method is not detailed. I understand it is extensively described in a manuscript under review in another journal, but there could be a summarized section in the present manuscript to briefly explain it. It would strengthen the results and the conclusions.

The paper in question has been published in an open access journal and the citation has been added. Based on the reviewer's comments, we have reviewed the methods section and made some improvements.

Similarly, the current manuscript does not describe the modern bird sample enough. As written, it is difficult to immediately see the phylogenetic diversity, number of species and number of modern bird individuals used in the study. A table should be added.

> As suggested, the modern bird sample has now been described more and an expanded Supplemental File has been added.

In addition, it lacks a section describing the limits of the study (problem with assessing a discrete ecology category to an animal, problem with the lack of certain bones, problem with position in the fossil, etc.), which would mitigate the impact of the results.

> We have reviewed the entire MS and ensured that the caveats of the method and of our interpretations have been provided and discussed.

Please, take note of the chronological comments and suggestions below.

Title

The title does not provide a clear view of the main result. As it is, it is too broad.

> We have adjusted the title.

Abstract

The abstract does not provide an accessible summary of the paper. There is no clear mention of the methods. The end part is a list of fossils with potential inferred ecologies, but the results and impact of the present study is not clearly stated.

> We have edited the abstract accordingly.

Line 43 page 2: "We interpret these foot data in the context of existing lines of evidence to help us better understand the evolutionary ecology of early theropod flyers."

Not informative sentence

> We have added a preceding sentence about methods that now makes this sentence informative.

Line 46 page 2: "more". Not clear to what it is compared, please rephrase.

> Rephrased as suggested.

Introduction

Line 63 page 3: "by comparing their toe pads, foot scales, claws and joints with living birds"

How do you compare? What methods? Please clarify

> We have added clarifying details to the text as suggested.

Line 69 page 4: "In modern birds, morphological differences in the scales and toe pads have been correlated to locomotory and feeding preferences"

How do they correlate? What feature is used and in what direction is the morphological is going?

> Thanks for bringing this up. This sentence should not have had a line break after it as the following paragraph is closely linked. We have joined the two passages together into one paragraph.

Line 107 page 5: "The morphometric dataset used here¹² was collected using the same measurements as 9,30."

Please briefly explain what the method is.

> We have done so as suggested.

Line 108 page 5: "This second dataset has a broader overall phylogenetic coverage, applies ecological categories to a broader range of taxa and uses bone-based landmarks that better apply to fossils"

Not clear what "this second dataset" is referring to. Are you talking about your dataset?

Please provide exact numbers to show how the dataset was improved

> We have clarified the text and provided exact numbers as suggested.

Line 116 page 6: "Non-raptorial perching birds have strongly curved claws¹²"

Not all non-raptorial perching birds do, in Dendrocolaptinae and Certhiidae the claw of the hallux is relatively straight while they climb upward on vertical surfaces.

Bock, W. J. and Miller, W. D. (1959). The scansorial foot of the woodpeckers, with comments on the evolution of perching and climbing feet in birds. American Museum novitates; no. 1931.

> Good point. We have adjusted our wording accordingly.

A reference about the different methods for assessing the shape of the claw should be added too: Tinius, A., & Patrick Russell, A. (2017). Points on the curve: an analysis of methods for assessing the

shape of vertebrate claws. *Journal of morphology*, 278(2), 150-169.

As well as a reference on the functional morphology of vertebrate claws.

Thomson, T. J. and Motani, R. (2021). Functional morphology of vertebrate claws investigated using functionally based categories and multiple morphological metrics. *Journal of Morphology Wiley* 282, 449–471.

> Additional references added as suggested.

Results

Line 151 page 7: “This collection revealed 12 specimens with exceptionally preserved toe pads, foot scales and claws belonging to the early flyers *Anchiornis*, *Archaeopteryx*, *Confuciusornis*, *Microraptor*, *Sapeornis*, and *Yanornis*”

What were the criteria to choose these specimens? Was phylogenetic diversity a criteria?

> Good point. We have added further details at the start of the results section.

The authors mentioned the good preservation, but later in the manuscript, they say: Line 165 page 8: “However, this observation is uncertain due to the poor preservation of the phalanges.”

Please clarify.

Line 165 is about *Ambopteryx*. This taxon is not mentioned in the list of specimens with well-preserved feet. We have clarified the foot preservation of *Ambopteryx* to avoid any confusion.

Discussion

Line 481 page 22: This paragraph should come earlier in the Discussion section, as it corresponds to the description of the overall story formed.

In general, the discussion should be revised to put forward the main results and contribution.

> We appreciate the comment, but have not made this change as we feel that this aspect is fine as is.

Images, Graphs and Data Tables

Figure 1. This figure should be improved. The legends corresponding to “arthrally” and “mesarthral” are too close to each other to clearly show the difference between the two. As it is something mentioned in the paper, this should be easily identified in the figure.

> Changed as suggested.

Figure 4 lacks a picture of the specimen.

> The specimen image is from a book that we do not hold the image copyright for. The citation is now noted in the figure legend (Figure 4 is now Figure 5).

A table with the dataset used for modern birds is missing (could be in supplementary materials)

> This has been added as suggested (Supplemental File 1).

Material and Methods

Line 709 page 40: "See 12 for additional details".

A list of the specimens with the name, family, collection number would be useful.

> The supplementary information is now referenced here as well.

Line 715 page 40: "the methodology of Wang et al. 96 based on Kaye et al. 36."

Same remark, having a sentence to explain the method would be useful.

> Detail added as suggested.

References

There are over 100 references, but important ones are missing. Please reconsider your list to make it more relevant to the paper.

We have reviewed the reference list and added new references include all of those suggested by the reviewers.

Reviewers' Comments:

Reviewer #1:

Remarks to the Author:

Dear authors,

the revision is throughout, and all my concerns have been addressed. I found no more issues. Well done!

Jens Lallensack

Reviewer #2:

Remarks to the Author:

I reviewed this manuscript twice previously, initially for a different journal. It has certainly improved with successive iterations.

I am satisfied with the changes made by the authors in this revision.

The manuscript can be published as is.

Denver Fowler.

Reviewer #3:

Remarks to the Author:

The authors provided a significant effort to take the three reviewers' comments into account. In my opinion, the revised version of the manuscript has greatly improved.

I am happy to see the published version of the methods, now cited in the manuscript. As well as an improved version of the supplementary materials. The availability of the measurements on both extinct and extant birds will ensure the repeatability of the study and strengthen the findings of the proposed study.

All my comments were considered and integrated in the revised version of the manuscript. I don't have any further comments to add.

Point-by-point Response to the Reviewers' Comments

Reviewer #1:

Dear authors,

the revision is throughout, and all my concerns have been addressed. I found no more issues. Well done!

Jens Lallensack

>> We are really glad that the revision is throughout and all of your concerns are addressed.

We wish to thank you again for your very helpful and constructive peer review.

Reviewer #2 (Remarks to the Author):

I reviewed this manuscript twice previously, initially for a different journal. It has certainly improved with successive iterations.

I am satisfied with the changes made by the authors in this revision.

The manuscript can be published as is.

Denver Fowler.

>> We are really glad that you are satisfied with the changes made and that you think the manuscript can be published as is.

We wish to thank you again for your very helpful and constructive peer review.

Reviewer #3 (Remarks to the Author):

The authors provided a significant effort to take the three reviewers' comments into account. In my opinion, the revised version of the manuscript has greatly improved.

I am happy to see the published version of the methods, now cited in the manuscript. As well as an improved version of the supplementary materials. The availability of the measurements on both extinct and extant birds will ensure the repeatability of the study and strengthen the findings of the proposed study.

All my comments were considered and integrated in the revised version of the manuscript. I don't have any further comments to add.

>> We are very happy that you think that the revised manuscript has greatly improved and for recognising the significant efforts made by our team to achieve this.

We are really glad that you are happy to see the published version of the methods now cited in the manuscript as well as the improved Supplementary Information. We are also very glad that you think that the repeatability of the study is now ensured by the edits made and that they have strengthened the findings of our study.

We are very pleased that you think that all of your comments were considered and integrated in the revised manuscript.

We note that you do not have any further comments to add.

We wish to thank you again for your very helpful and constructive peer review.